# HIV-1 cell-to-cell spread overcomes the virus entry block of non-macrophage-tropic strains in macrophages

**Mingyu Han**[1,2,3☯], **Vincent Cantaloube-Ferrieu**[4☯], **Maorong Xie**[1,2,3¤], **Marie Armani-Tourret**[4], **Marie Woottum**[1,2,3], **Jean-Christophe Pagès**[5], **Philippe Colin**[4], **Bernard Lagane**[ID][4‡]*, **Serge Benichou**[ID][1,2,3‡]*

**1** Institut Cochin, Inserm U1016, Paris, France, **2** CNRS, UMR8104, Paris, France, **3** Université de Paris, Paris, France, **4** Infinity, Université de Toulouse, CNRS, INSERM, UPS, Toulouse, France, **5** Institut RESTORE, Université de Toulouse, CNRS U-5070, EFS, ENVT, Inserm U1301, Toulouse, France

☯ These authors contributed equally to this work.
¤ Current address: Division of Infection and Immunity, University College London, London, United Kingdom
‡ These authors are joint senior authors on this work.
* bernard.lagane@inserm.fr (BL); serge.benichou@inserm.fr (SB)

**Data Availability Statement:** All relevant data are within the manuscript and its Supporting Information files.

## Abstract

Macrophages (MΦ) are increasingly recognized as HIV-1 target cells involved in the pathogenesis and persistence of infection. Paradoxically, *in vitro* infection assays suggest that virus isolates are mostly T-cell-tropic and rarely MΦ-tropic. The latter are assumed to emerge under CD4+ T-cell paucity in tissues such as the brain or at late stage when the CD4 T-cell count declines. However, assays to qualify HIV-1 tropism use cell-free viral particles and may not fully reflect the conditions of *in vivo* MΦ infection through cell-to-cell viral transfer. Here, we investigated the capacity of viruses expressing primary envelope glycoproteins (Envs) with CCR5 and/or CXCR4 usage from different stages of infection, including transmitted/founder Envs, to infect MΦ by a cell-free mode and through cell-to-cell transfer from infected CD4+ T cells. The results show that most viruses were unable to enter MΦ as cell-free particles, in agreement with the current view that non-M-tropic viruses inefficiently use CD4 and/or CCR5 or CXCR4 entry receptors on MΦ. In contrast, all viruses could be effectively cell-to-cell transferred to MΦ from infected CD4+ T cells. We further showed that viral transfer proceeded through Env-dependent cell-cell fusion of infected T cells with MΦ targets, leading to the formation of productively infected multinucleated giant cells. Compared to cell-free infection, infected T-cell/MΦ contacts showed enhanced interactions of R5 M- and non-M-tropic Envs with CD4 and CCR5, resulting in a reduced dependence on receptor expression levels on MΦ for viral entry. Altogether, our results show that virus cell-to-cell transfer overcomes the entry block of isolates initially defined as non-macrophage-tropic, indicating that HIV-1 has a more prevalent tropism for MΦ than initially suggested. This sheds light into the role of this route of virus cell-to-cell transfer to MΦ in CD4+ T cell rich tissues for HIV-1 transmission, dissemination and formation of tissue viral reservoirs.

**Funding:** This work was supported in part by Inserm, CNRS, and the University Paris-Descartes. It was also funded by grants from the Agence Nationale de Recherche sur le SIDA et les Hépatites Virales (ANRS). M.H. and M.X. were supported by grants from the China Scholarship Council (CSC). V.C.-F. and M.W. were supported by a grant from ANRS. M.A.-T was funded by a grant from Sidaction. The funders play any role in the study design, data collection and analysis, decision to publish, or preparation of the manuscript.

**Competing interests:** The authors have declared that no competing interests exist.

## Author summary

Understanding how HIV-1 hijacks the functions of immune cells to promote viral spreading remains a challenge in the fight against infection. MΦ are ubiquitous tissue-resident cells, involved in tissue homeostasis and immunity. In HIV-1 infection, along with CD4+ T lymphocytes, MΦ serve as vectors for virus dissemination and as viral reservoirs, impeding HIV-1 eradication. However, the mechanisms of their infection remain incompletely understood. A paradox is that infected MΦ are found in a large range of tissues whereas *in vitro* cellular tropism assays indicate that only a limited number of HIV-1 isolates can enter MΦ. We hypothesized that these assays, which evaluate infection using cell-free viruses, might not fully reflect the modes of MΦ infection in patients. We report here that virus cell-to-cell transfer through cell-cell fusion with infected CD4+ T cells is a very effective means of infecting MΦ, even with virus isolates characterized as non-macrophage tropic in cell-free infection. This intercellular viral transfer is facilitated by enhanced interactions between the HIV-1 envelope glycoproteins and cellular entry receptors. We propose that MΦ infection through viral transfer from infected CD4+ T cells impacts different aspects of the pathophysiology of HIV-1 infection, renewing our understanding of the role of MΦ in HIV-1 pathogenesis and persistence.

## Introduction

In addition to CD4+ T lymphocytes (CD4TL), cells of the macrophage (MΦ)-monocyte lineage are HIV-1 target cells that play a central role in the pathophysiology of HIV-1 infection leading to AIDS. MΦ contribute to the dissemination of the virus into different tissues such as the liver, digestive and genital tracts, lungs, central nervous system (CNS) and lymphoid organs [1–8]. Together with latently-infected CD4 T cells, they also act as cellular reservoirs for the virus, which represent a major obstacle to its eradication and can contribute to persistence of adverse effects in infected patients despite effective antiretroviral therapy [9–11]. For instance, residual infection of perivascular MΦ and microglial cells in the CNS is estimated to be the cause of chronic brain inflammation and neurocognitive disorders experienced by half of the patients under therapy [12]. This persistence in MΦ may contribute to viral rebound following treatment interruption [13,14].

However, the mechanisms leading to MΦ infection are still incompletely understood. Compared to CD4TL, MΦ are renowned to have a lower susceptibility to HIV-1. Most HIV-1 variants described to infect MΦ are R5-tropic, *i.e.* their envelope glycoprotein (Env) uses CC chemokine receptor 5 (CCR5) as a coreceptor of CD4 for viral entry [15,16]. Infection of MΦ by viruses using CXC chemokine receptor 4 (CXCR4) as a coreceptor (X4- and R5X4-tropic viruses) is less frequent, although reported [4,17]. However, even for R5 viruses, only a small fraction can effectively infect MΦ in cell-free virus infection assays [16]. A paradigmatic example of this is represented by transmitted/founder (T/F) HIV-1 isolates, which are typically R5-tropic but are very inefficient to infect MΦ [18,19]. These observations have led to the classification of HIV-1 isolates according to their viral tropism (the type of coreceptor used) and cellular tropism as X4 T-tropic, R5 T-tropic, and M-tropic viruses [16]. The first two categories are thought to represent the majority of viruses, often also termed non-M-tropic viruses. The reasons invoked to explain why most HIV-1 isolates are non-M-tropic viruses are multiple. MΦ express a number of restriction factors that are part of the host innate immune response and strongly inhibit HIV-1 replication at different post-entry stages of the life cycle. These include SAMHD1 and members of the APOBEC3, TRIM and IFITM families [15,20,21].

However, the fact that most HIV-1 isolates fail to infect MΦ is primarily due to their inability to enter these cells [16]. This has been generally explained by a lower efficiency of interaction with entry receptors on MΦ than on CD4TL. In particular, several studies have proposed that most viruses are inefficient at entering MΦ because these cells express lower CD4 density than CD4TL [22,23]. Supporting this view, many HIV-1 isolates that effectively infect MΦ, most isolated from the CNS [5], are featured by enhanced Env/CD4 interactions [22–27]. In some cases, the M- or non-M-tropism has been linked to the properties of Env interactions with CCR5 [28,29] or CXCR4 [30]. Accordingly, we recently showed that entry of some blood-derived isolates into MΦ was not related to enhanced interaction with CD4 but to Env's ability to recognize particular CCR5 conformations at the cell surface [31].

Hence, the current view is that HIV-1 isolates equipped to infect MΦ are uncommon [16]. Genuine M-tropic viruses are thought to emerge through adaptation to body niches where MΦ or related cells are present in a context of CD4+ T-cell paucity, in particular at the late stage of infection when the CD4 T-cell count declines. This remains paradoxical regarding the presence of HIV-1-infected MΦ in a wide range of tissues [1–8] and their capacity to promote systemic dissemination of the virus [3,32,33]. A possible explanation for this apparent paradox is that the methods used to classify HIV-1 MΦ tropism do not recapitulate *in vivo* modes of infection. Macrophage-tropism of viruses has been most often defined by *in vitro* infection experiments of blood monocyte-derived macrophages (MDMs) with cell-free viral particles. Because differences are often observed in the ability of viruses to infect MDMs from one donor to another, the HEK 293-derived Affinofile cell line in which CD4 and CCR5 expression levels can be independently manipulated has also been used as MDM surrogates to discriminate between M- and T-tropic viruses [22,23]. In general, M-tropic viruses but not non-M-tropic viruses are able to infect Affinofile cells expressing low levels of CD4 as in macrophages. However, it is now clearly established that enveloped viruses such as HIV-1 can infect target cells not only by a cell-free mode but also through cell-to-cell transmission involving direct contacts between infected cells and uninfected target cells [34–36]. This mode of infection that can result from several mechanisms, the most paradigmatic involving formation of a virological synapse between virus-donor and target cells, is considered more efficient than cell-free infection [37–39]. Cell-to-cell transmission may be the primary route of HIV-1 infection and spreading *in vivo*, particularly in tissues with high HIV-1 target cell density [40] and when barriers render target cells poorly susceptible to cell-free infection, e.g. under conditions of low virus binding efficiency [41]. In this respect, previous works showed that MΦ could be infected *via* phagocytosis of and/or contacts with infected CD4+ T cells, even by weakly M-tropic viruses [7,32,42–44]. Recently, we revealed that MΦ and related myeloid cells can be infected from infected T cells by a two-step Env-dependent cell-cell fusion process leading to the formation of highly virus-productive multinucleated giant cells (MGCs) [45–47]. Contacts between target MΦ and infected T cells involve sequential interactions between the Env surface subunit gp120 exposed at the T-cell surface and CD4 and the coreceptor expressed on MΦ. Then, the Env membrane subunit gp41 triggers heterotypic cell-cell fusion for virus transfer into MΦ. Then, cell-cell fusion of the newly formed T cell/MΦ fused cells with uninfected neighboring MΦ ultimately leads to massive virus dissemination. Importantly, this cell-cell fusion process of MΦ infection is not restricted by the SAMHD1 restriction factor [47], suggesting that this mode of HIV-1 spreading in MΦ could be more efficient than cell-free infection. Therefore, cell-to-cell transmission of HIV-1 to MΦ could represent a dominant mode for virus dissemination in myeloid cells *in vivo*. Accordingly, infection of myeloid cells through cell-cell fusion has long been suggested *in vivo* by the presence of MΦ- and DC-derived MGCs in the brain and lymphoid tissues of infected individuals or in monkey models, even under effective suppressive antiretroviral therapy [48–55].

The goal of this study was to investigate whether virus isolates, initially defined as non-M-tropic on the basis of cell-free infection assays, could efficiently infect MΦ through cell-cell fusion of infected T cells with MΦ targets. Using a large panel of viruses expressing primary Envs with CCR5 and/or CXCR4 usage and from different stages of infection, we show that all viruses can be transferred from infected T cells to MΦ ultimately leading to the formation of productively infected MGCs. These results indicate that HIV-1 infection by cell-cell fusion can overcome barriers that usually restrict infection of MΦ by cell-free viral particles. Indeed, we show that infection of MΦ through cell-to-cell virus transfer is linked to enhanced Env interactions with CD4 and the coreceptor, thereby making viral entry less dependent on the expression level of both receptors at the surface of MΦ. These data support a model in which HIV-1 primary isolates have a wider tropism for MΦ than suggested by *in vitro* assays using cell-free viral particles. This further highlights the role of MΦ in virus transmission and dissemination to tissues influencing the pathophysiology of HIV-1 infection.

## Results

### Cell-free HIV-1 particles expressing primary Env inefficiently infect macrophages *in vitro*

As a first step, we characterized the capacity of a panel of viruses to infect MDMs when used as cell-free viral particles (**Fig 1**). We used replication competent, NL4-3-derived luciferase reporter viruses pseudotyped with R5 or X4 Envs of biological virus clones isolated from peripheral blood mononuclear cells (PBMCs) of patients at the chronic or AIDS stage of infection [31,56]. We compared the capacity of different amounts of the R5 viruses, generated from HEK 293T cells, to infect PHA/IL-2-activated primary CD4TL from healthy blood donors (**Fig 1A**). Then, equivalent infectious doses of viruses (35,000 RLU) were used to inoculate MDMs from the same (*i.e.* autologous MDMs) or unrelated individuals (**Fig 1B**). Parallel infections with the prototypic non-M- and M-tropic viruses JR-CSF and JR-FL, respectively, served as controls. Results indicate that infectivity in MDMs greatly varies between these R5 viruses, but in a manner that does not correlate with infectivity in CD4TL. Similar to JR-FL, only two viruses consistently infected MDMs, R5-13 and R5-34, whereas the others did not or only marginally (R5-58). R5-34 displayed the weakest infectivity in CD4TL, but it was most effective at infecting MDMs. Of note, all viruses efficiently infected CD4TL, and infectivities between viruses measured in CD4TL varied less than in MDMs (**Fig 1A**), in agreement with the notion that blood-derived Envs are more commonly T-tropic than M-tropic [23]. Close examination of the results indicates, however, that the infectivity in MDMs greatly varies with the donor (indicated by the color code of the dots in **Fig 1B**), as previously reported [23,31], in a virus type-dependent manner (**Fig 1B**). For instance, compared to JR-FL, the M-tropic R5-34 infected MDMs with higher or lower efficacy depending on the donor. Also, viruses mostly unable to infect MDMs (e.g. R5-25) occasionally infected MDMs from some donors.

We next investigated whether the differences of these viruses to infect MDMs were related to differences in efficiency of entry or post-entry steps. For this, we incorporated the β-lactamase (BLaM)-Vpr fusion protein into some of the above viruses and then analyzed their capacity to infect CD4TL (**Fig 1C**) and fuse with MDMs pre-loaded with the fluorescent BLaM substrate CCF2-AM (**Fig 1D**). Fusion was quantified by flow cytometry analysis of fluorescence changes in MDMs upon CCF2 cleavage by BLaM, as detailed previously [31]. As above, MDMs were inoculated with equivalent amounts of BLaM-Vpr-containing viruses (35,000 or 100,000 RLU in CD4TL, dashed lines in **Fig 1C**). Results showed that the R5 viruses able to infect MDMs (see **Fig 1B,** M-tropic viruses R5-13, R5-34 and JR-FL) also more frequently fused with MDMs, compared to non-M-tropic viruses, indicating that the capacity of viruses

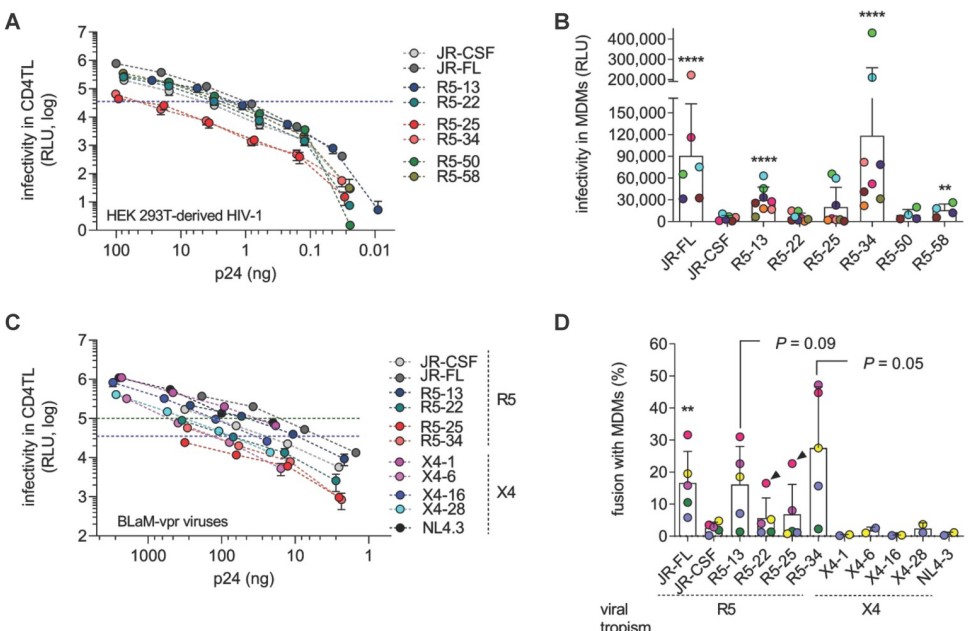

**Fig 1. Cell-free T-tropic HIV-1 particles enter and infect MDMs at low efficiency. A)** The infectivity of cell-free viruses pseudotyped with the indicated R5 Envs was measured in PHA/IL-2-activated CD4TL as a function of the Gag p24 content in serially diluted supernatants from HEK 293T producer cells. Results (means ± SEM of three independent experiments) show the luciferase activity in the lysates of infected cells, expressed as relative light units (RLU), measured 48 h post-inoculation. The dashed line depicts the infectious dose of the different viruses that was subsequently used to infect MDMs (35,000 RLU equivalents). **B)** Infectivity of the different viruses in MDMs. Each point represents the average infectivity of 3 replicates measured with the MDMs from one donor. MDMs from different donors were used, each depicted by a color code. For each virus, infectivity is expressed as means ± SD of the different measurements that were carried out with the MDMs from the different donors. **C)** Infectivity of BLaM-vpr-containing viruses pseudotyped with the indicated R5 or X4 Envs in activated CD4TL, expressed as in panel (**A**). BLaM-vpr containing viruses were concentrated by centrifugation and then diluted to different p24 concentrations before being exposed to target cells. The dashed lines indicate the two infectious doses of viruses that were subsequently used to infect MDMs (35,000 or 100,000 RLU equivalents). **D)** Percentage of viral fusion with MDMs, expressed as means ± SD of experiments carried out with MDMs from distinct donors. Each donor is depicted by a color code. X4 viruses were used at an infectious dose of 100,000 RLU equivalents, R5 viruses at an infectious dose of 35,000 RLU (except MDMs depicted by purple dots which were infected with 100,000 RLU). Strawberry-colored symbols (arrowheads) represent the results obtained with one donor's MDMs that were also used in experiments shown in panel (**B**). Statistics were run using the Mann-Whitney U-test: **, $P < .01$; ****, $P < .0001$.

to infect MΦ was largely dependent on their ability to enter these cells. As observed in the cell-free infection assays (**Fig 1B**), the proportion of MDMs that support virus fusion was critically influenced by the blood donor. Non-M-tropic viruses also occasionally fused with MDMs of some donors, albeit less efficiently, compared to M-tropic viruses. In one case where MDMs from the same donor were used in both infection and fusion experiments (strawberry-colored symbols depicted with an arrowhead in **Fig 1D**), the non-M-tropic viruses R5-22 and R5-25 could fuse with the MDMs from this specific donor, but without leading to productive infection (**Fig 1B**). This feature is consistent with previous works and could be explained by a post-entry restriction [57].

Finally, we performed fusion experiments using viruses pseudotyped with primary X4 Envs or Env of the X4 strain NL4-3 known to be unable to promote macrophage infection [58]. These viruses consistently failed to enter MDMs (**Fig 1D**), even though they were, as R5 viruses, effective at infecting CD4TL (**Fig 1C**). These results therefore support the idea that M-tropic viruses are rare among viruses using CXCR4 as coreceptor [15,16].

Altogether, the results reported in **Fig 1** indicate that there is a great variability in the ability of primary HIV-1 strains to infect MΦ using cell-free viral particles. Among the viruses we have analyzed, fusion assays indicated that several R5 viruses and all X4 viruses were primarily blocked at the viral entry step. Even for the viruses successfully infecting MΦ, variations in entry efficiency and infectivity were depending on the donor.

## Both R5 M- and non-M-tropic HIV-1 isolates are transferred to macrophages from infected T cells through Env-mediated cell-cell fusion

We then investigated the ability of the above R5 Env-pseudotyped viruses to infect MΦ by cell-to-cell transfer from infected CD4+ T cells. For this purpose, replicative viruses pseudotyped with the vesicular stomatitis virus envelope G glycoprotein (VSVg) were produced as previously [45–47], and used to infect human CD4+ Jurkat T cells. Forty-eight hours later, infected Jurkat cells, which only express HIV-1 Envs at their surface, were used as virus-donor cells in coculture experiments for 6 or 24 h with MDMs used as target cells, as detailed previously [45,47] (schematized in **Fig 2A**). Then, Jurkat cells in suspension were eliminated by extensive washes, and the adherent MDMs were harvested, stained for the intracellular viral Gag p24 antigen and analyzed by flow cytometry. For comparison, MDMs derived from the same donors were also exposed for 6 or 24 h to cell-free virus-containing supernatants from infected Jurkat cells. As expected, viruses pseudotyped with JR-FL Env or with the Envs defined as M-tropic (Envs R5-13, -34 and -58) were efficiently transferred from infected T cells to MDMs (**Fig 2B** and **2C**, CT). Of note, the percentage of Gag+ MDMs varied greatly depending on the donor, as observed in MDM infection by cell-free HIV-1 (see **Fig 1**). In contrast, no Gag + MDMs were detected when using supernatants from infected Jurkat cells (**Fig 2B** and **2C**, CF), suggesting that Gag transfer to MDMs depends on cell-to-cell contacts and not on cell-free viruses. Interestingly, significant transfer of viral material was detected in MDMs as soon as 6 h of coculture with Jurkat cells infected with viruses pseudotyped with non-M-tropic Envs (Envs R5-22, -25 and -50, as well as JR-CSF) (**Fig 2B**). However, transfer was often less efficient than when using M-tropic viruses, but these differences between M- and non-M-tropic viruses tended to decrease when the duration of coculture was extended to 24 h (**Fig 2C**). We next performed luciferase assay to further confirm that cell-to-cell transfer of both M- and non-M-tropic viruses from infected T cells resulted in infection of the target MDMs. MDMs were cocultured with infected Jurkat cells or exposed to supernatants from infected Jurkat cells for 24 h, then extensively washed and cultured for 48 additional hours before quantification of the luciferase activity (**Fig 2D**). Compared to the prototypic M-tropic virus JR-FL, all viruses, including the non-M-tropic, expressed similar luciferase activity when MDMs were initially cocultured with infected Jurkat cells. This activity was almost totally inhibited through treatment of the target MDMs with the Env-binding fusion inhibitor T20. In contrast, no significant luciferase activity was detected in MDMs exposed to the supernatants of infected Jurkat cells, except for the JR-FL-infected Jurkat cell supernatant. Together, these results indicate that the R5 M-tropic isolates, but also those initially defined as non-M-tropic viruses by cell-free infection assay, can be efficiently transferred to MΦ from infected T cells through cell-to-cell contacts, leading to efficient infection of the target MΦ.

We previously showed that CD4+ T cell-mediated cell-to-cell transfer of laboratory-adapted M-tropic HIV-1 strains to MΦ was mainly mediated by a two-step cell-cell fusion process leading to the formation of productively infected MGCs [45,47]. Using immunofluorescence analysis, we thus investigated whether transfer of the R5 M- or non-M-tropic viruses involved cell-cell fusion of infected T cells with target MDMs. Infected Jurkat cells were pre-loaded with the intravital CellTracker dye for staining of nuclei and then cocultured with

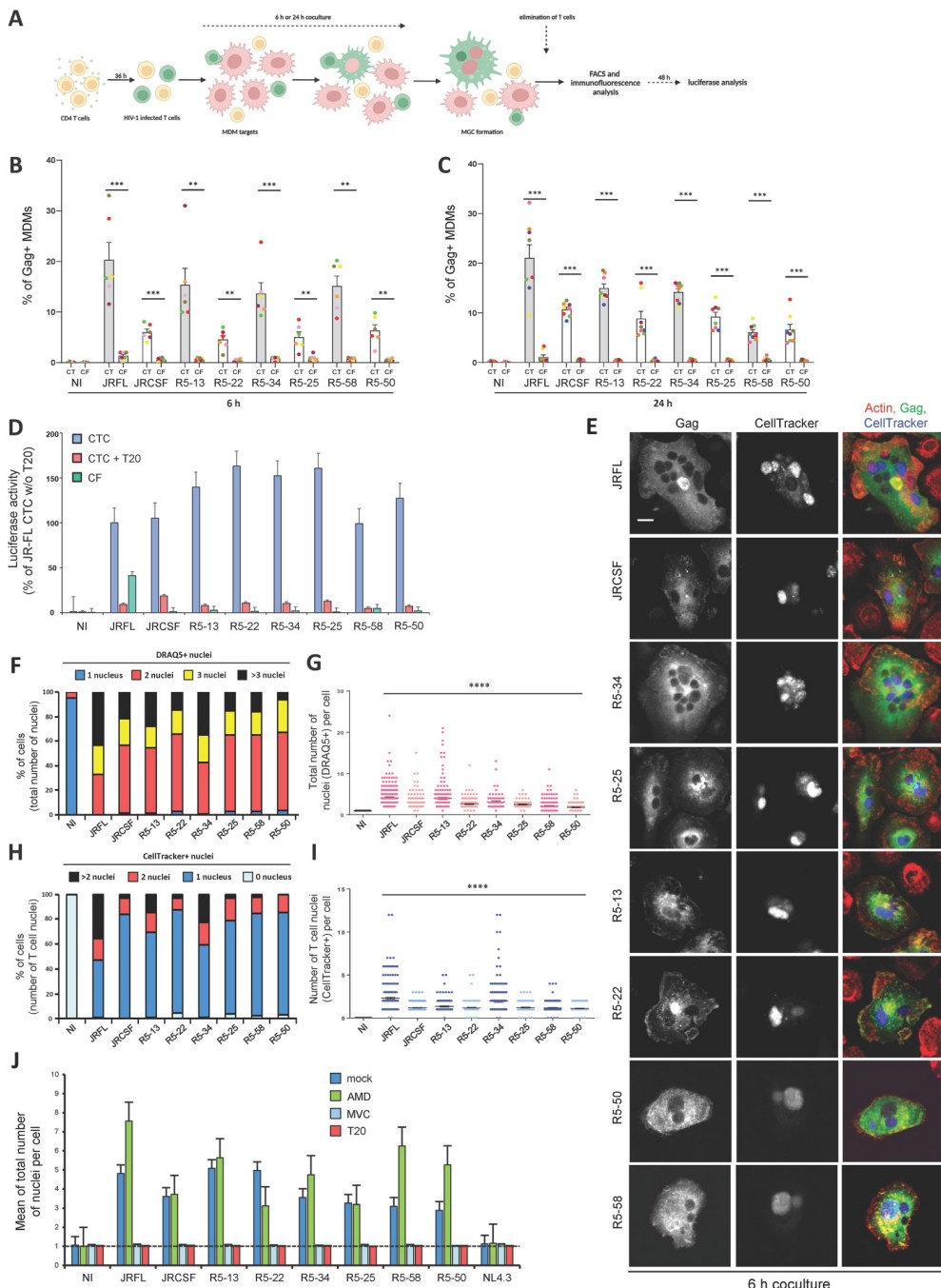

**Fig 2. Cell-to-cell transfer to macrophages of R5 M- and non-M-tropic HIV-1 through cell-cell fusion with infected T cells. A)** Experimental design for HIV-1 cell-to-cell transfer from infected CD4+ T cells to MDMs. **B, C)** MDMs were exposed to cell-free (CF) virus-containing supernatants of Jurkat cells infected with viruses pseudotyped with the indicated CCR5-using Envs or cocultured (CT) with Jurkat cells infected with the corresponding M- (grey bars) or non-M-tropic (white bars) viruses. The percentage of Gag+ MDMs was then determined by flow cytometry 6 (**B**) or 24 (**C**) h later. The results show the percentage of Gag+ MDMs. Non-infected MDMs or MDMs cocultured with non-infected Jurkat cells were used as negative controls (NI) of cell-free or cell-to-cell infection, respectively. Each dot corresponds to MDMs from an individual donor. **D)** MDMs were infected by cell-free (green bars) or cell-to-cell viral transfer in the presence (red bars) or absence (blue bars) of T20 for 24 h. After elimination of the cell-free virus inoculum or infected Jurkat cells, MDMs were cultured for 48 more hours before quantification of the luciferase activity. The results are expressed as the percentage of the luciferase activity measured for the JRFL Env-pseudotyped viral clone. **E-J)** Jurkat cells infected with the indicated Env-pseudotyped virus clones and prelabeled with CellTracker were co-cultured for 6 (**E-I**) or 24 (**J**) h with MDMs. After elimination of Jurkat cells, MDMs were stained with anti-

Gag, phalloidin (actin) and DRAQ5 (nuclei), and analyzed by confocal microscopy (scale bar, 10 μm). Representative images are shown in (**E**). The number of T cell nuclei (CellTracker+) as well as the total number of nuclei (DRAQ5+) per Gag+ MDM were quantified from images on at least 200 cells. MDMs cocultured with non-infected Jurkat cells were used as negative controls (NI). In **F**), results are expressed as the percentages of Gag+ MDMs with 1, 2, 3 or more than 3 DRAQ5(+) nuclei. In **G**), results are expressed as the number of DRAQ5(+) nuclei per MDM initially cocultured with Jurkat cells infected with M-tropic (dark pink) or non-M-tropic (light pink) viruses; each dot corresponds to 1 cell. In **H**), results are expressed as the percentages of Gag+ MDMs with 0, 1, 2, or more than 2 CellTracker(+) nuclei. In **I**), results are expressed as the number of CellTracker(+) nuclei per Gag+ MDM initially cocultured with Jurkat cells infected with M-tropic (dark blue) or non-M-tropic (light blue) viruses; each dot corresponds to 1 cell. In **J**), results are expressed as the means of total nucleus number (DRAQ5+) per Gag+ MDM pretreated or not (dark blue bars, mock) with AMD3100 (green, AMD), maraviroc (light blue, MVC) or T20 (red bars), and then cocultured with infected Jurkat cells. In **B-D**, and **J**), the results represent means of at least 6 independent experiments performed with MDMs of at least 6 different donors; error bars represent 1 standard error of the mean (SEM). Statistical significance was determined using the Mann-Whitney U-test (**, $P < 0.01$; ***, $P < 0.001$; ****, $P < .0001$). In **E-I**), the results shown are representative of at least 4 independent experiments performed with MDMs from at least 4 different donors; horizontal bars in (**G**) and (**I**) represent means +/- SEM.

MDMs for 6 or 24 h. MDMs were then stained for intracellular Gag p24, and the DRAQ5 dye was used to stain all nuclei before observation by fluorescence confocal microscopy (**Figs 2E–2I** and **S1**). After 6 h of coculture with infected T cells, all Gag+ MDMs exhibited a strong and diffuse cytoplasmic Gag staining (**Fig 2E**), and contained at least 2 nuclei (**Fig 2F** and **2G**) including 1 CellTracker+ dye-stained nucleus signing a T cell origin (**Fig 2H** and **2I**). The number of nuclei per MDM and the proportion of MGCs formed increased further upon coculture to 24 h (**S1 Fig**). These results confirm that cell-cell fusion is the main mechanism for virus transfer from infected T cells to MDMs, even when T cells are infected with non-M-tropic viruses. As shown in **Fig 2J**, cell-cell fusion of infected T cells with MDMs and MGC formation were totally inhibited by the CCR5 ligand maraviroc (MVC) or the fusion inhibitor T20, but not by the CXCR4 antagonist AMD3100 (AMD). This confirms that cell-to-cell transfer of R5 viruses is mediated by interactions between the viral envelope and the CCR5 coreceptor expressed at the cell surface of infected T cells and MΦ, respectively.

Together, these results show that M-tropic and non-M-tropic CCR5-using viruses are able to mediate cell-cell fusion between infected T cells and MΦ leading to efficient cell-to-cell virus spreading.

## Infected T cells transfer of non-M-tropic CXCR4-using HIV-1 to macrophages through cell-cell fusion

Because CXCR4-using viruses are considered as T-tropic and rarely M-tropic [15,16], as shown using cell-free virus infection assays (**Fig 1**), we interrogated whether they could be transferred from infected T cells to MΦ by cell-cell fusion. Jurkat cells were infected with the CXCR4-using viruses described above, and then cocultured for 6 or 24 h with MDMs before quantification of the percentage of Gag+ MDMs by flow cytometry. Viruses pseudotyped with Env from the dual(R5X4)-tropic isolate 89.6, known to replicate in MDMs [59], were also included in this analysis. As shown in **Fig 3A and 3B**, all viruses were transferred from infected T cells to MDMs (CT), whereas no Gag+ MDMs was detected using supernatants from infected Jurkat cells (CF). Luciferase assays performed 48 h after the coculture confirmed that cell-to-cell transfer of these CXCR4-using viruses led to infection of the target MDMs, which was significantly inhibited by T20 (**Fig 3C**).

Immunofluorescence analysis confirmed that Jurkat cells infected with the X4 Env-pseudo-typed viruses fused with MDMs and led to the formation of Gag+ MGCs (**Figs 3D–3F,** and **S2**) containing at least 1 CellTracker+ nucleus coming from infected T cells (**Fig 3G and 3H**). In contrast, no significant cell-cell fusion was observed when MDMs were cocultured with T

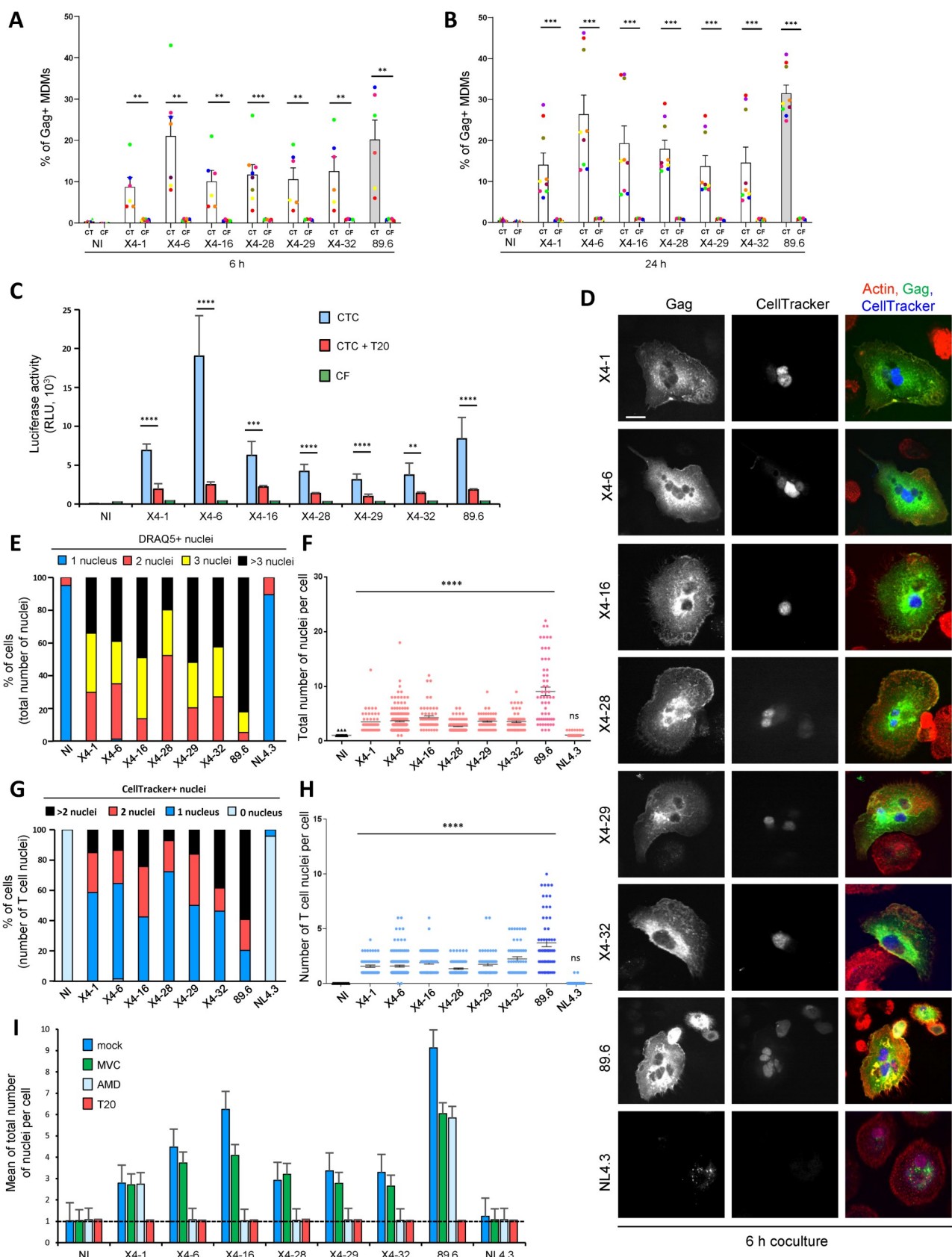

**Fig 3. Cell-to-cell transfer to macrophages of CXCR4-using HIV-1 through cell-cell fusion with infected T cells. A** and **B),** MDMs were exposed to cell-free (CF) virus containing-supernatants of Jurkat cells infected with viruses pseudotyped with the indicated CXCR4-using Envs or cocultured (CT) with the corresponding infected Jurkat cells for 6 h (**A**) or 24 h (**B**). The virus pseudotyped with Env from the dual X4/R5 M-tropic isolate 89.6 (grey bars) was used as a positive control. The percentage of Gag+ MDMs was then determined by flow cytometry. Each dot corresponds to MDMs from an individual donor. Non-infected MDMs or MDMs cocultured with non-infected Jurkat cells were used as negative controls (NI) of CF or CTC infection, respectively. **C)** MDMs were infected by cell-free (CF, green bars) or by cell-to-cell (CTC) virus transfer in the presence (red bars) or absence (blue bars) of T20 for 24 h as previously. After elimination of the cell-free virus inoculum or Jurkat cells, MDMs were cultured for 48 more hours before quantification of the luciferase activity. **D-I)** Jurkat cells infected with the indicated virus clones and prelabeled with CellTracker were co-cultured for 6 h (**D-H**) or 24 h (**I**) with MDMs. After elimination of Jurkat cells, MDMs were stained with anti-Gag, phalloidin (actin) and DRAQ5, and analyzed by confocal microscopy. Representative images are shown in (**D**) (scale bar, 10 μm). The number of T cell nuclei (CellTracker+) as well as the total number of nuclei (DRAQ5+) per Gag+ MDM were quantified from images on at least 200 cells. MDMs cocultured with non-infected Jurkat cells were used as negative controls (NI). In **E)**, results are expressed as the percentages of Gag+ MDMs with 1, 2, 3 or more than 3 DRAQ5(+) nuclei. In **F)**, results are expressed as the number of DRAQ5(+) nuclei per Gag+ MDM; each dot corresponds to 1 cell. In **G)**, results are expressed as the percentages of Gag+ MDMs with 0, 1, 2, or more than 2 CellTracker(+) nuclei. In **H)**, results are expressed as the number of CellTracker(+) nuclei per Gag+ MDM; each dot corresponds to 1 cell. In **I)**, results are expressed as the means of total nucleus number (DRAQ5+) per Gag+ MDM pretreated or not (dark blue bars, mock) with AMD3100 (green bars, AMD), maraviroc (light blue bars, MVC) or T20 (red bars) and then cocultured with infected Jurkat cells. In **A-C**, and **I)**, the results represent means of at least 6 independent experiments performed with MDMs of at least 6 different donors; error bars represent 1 SEM. Statistical significance was determined using the Mann-Whitney U-test (**, $P < 0.01$; ***, $P < 0.001$; ****, $P < 0.0001$). In (**D-H**), the results shown are representative of at least 4 independent experiments performed with MDMs from at least 4 different donors; horizontal bars in (**F**) and (**H**) represent means +/- 1 SEM.

cells infected with viruses pseudotyped with NL4.3 Env (**Figs 3D–3H,** and **S2**), and only a dotted Gag staining was observed in some mononucleated MDMs even after 24 h of coculture (**Figs 3D** and **S2**). These results are reminiscent of previous works using cell-free viral particles showing that fusogenicity of laboratory-adapted X4 Envs, such as NL4.3 Env, with myeloid cells is lower compared to primary X4 Envs [17,60]. Finally, to confirm the use of CXCR4 to mediate cell-cell fusion, the formation of Gag+ MGCs was analyzed in the presence of AMD3100, MVC or T20 (**Fig 3I**). With the exception of viruses pseudotyped with Env X4-1, cell-cell fusion between MDMs and Jurkat cells infected with viruses harboring the 5 other primary Envs (X4-6, -16, -28, -29 and -32) was totally blocked by AMD3100 and T20, but not by MVC (**Fig 3I**), confirming the use of CXCR4 for cell-cell fusion and MGC formation. As expected, virus transfer and formation of MGCs mediated by 89.6 Env were inhibited by a mixture of AMD3100 and MVC (**S3 Fig**), consistent with the ability of this Env to use both CCR5 and CXCR4. Similarly, cell-to-cell transfer of viruses pseudotyped with Env X4-1 was inhibited by a mixture of AMD3100 and MVC, but not by AMD3100 or MVC alone (**Figs S3** and **3I**), indicating that Env X4-1 can use both CXCR4 and CCR5 for cell-cell fusion and MGC formation.

These results show that CXCR4-using Envs, characterized as non-M-/T-tropic by cell-free infection assay, mediate cell-cell fusion and MGC formation supporting viral spread in MΦ.

## HIV-1 particles produced from T cells show low ability to infect macrophages

We next interrogated how MDMs could be efficiently infected by non-M-tropic viruses through virus cell-to-cell transfer from infected T cells but not with cell-free viruses. Because the experiments reported in **Fig 1** have been performed using viruses produced in HEK 293T cells, we wondered whether the efficient transfer of viruses from T cells to MDMs could be due to the fact that viruses produced in T cells have characteristics favouring M-tropism. To investigate whether the cellular tropism of viruses is dependent upon the cells from which they originate, we first compared the capacity of similar quantities (10 ng of Gag p24) of the R5 Env-pseudotyped viruses, produced either in HEK 293T or Jurkat cells, to infect primary CD4TL from distinct donors. Of note, both virus types expressed similar levels of Env on their surface, as shown by western blot analysis of the amounts of gp120 and p24 into JR-FL Env-

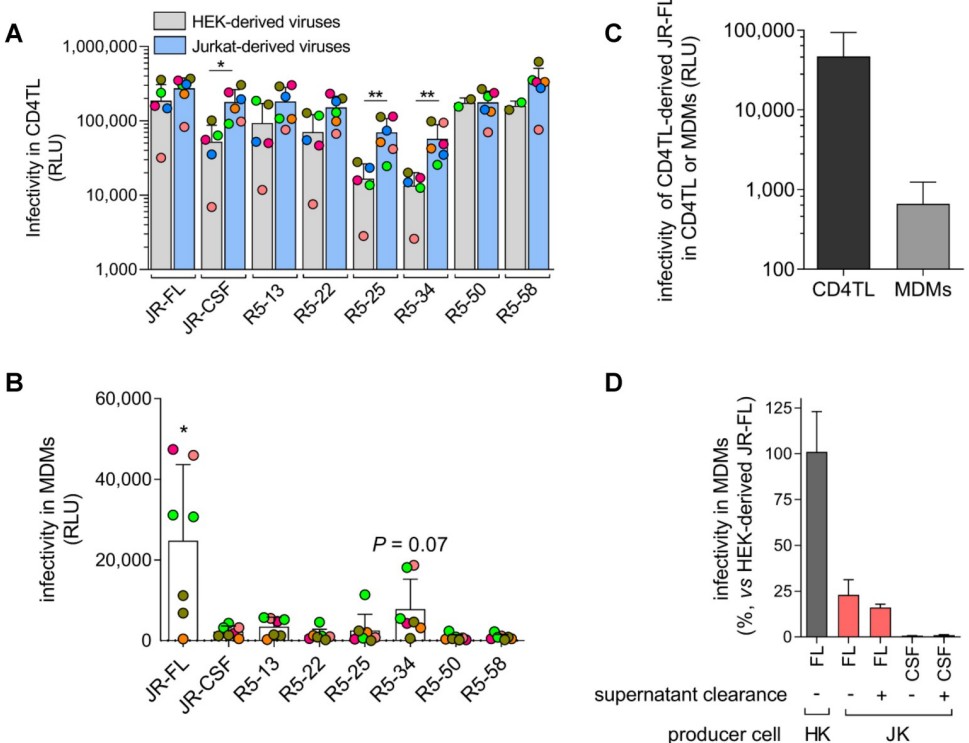

**Fig 4. HIV-1 particles produced in T cells have a weak ability to infect MDMs. A)** Similar amounts (10 ng of Gag p24) of R5 Env-pseudotyped viruses produced in Jurkat cells (JK) or HEK 293T cells (HEK) show comparable capacity to infect primary CD4TL. Each dot represents the average infectivity measured with the CD4TL from one donor. Each color represents a particular donor. Results expressed as RLU in the lysates of infected CD4TL are means ± SD. **B)** Infectivity of the different JK-derived viruses in MDMs. Experiments were carried out as in **Fig 1B** related to HEK-derived viruses. In particular, the same infectious doses of viruses as in **Fig 1B** were used (35,000 RLU). MDMs from different donors were used, each depicted by a color code (note that MDMs from the same individual may have been used twice in some cases). Results are means ± SD. **C)** JR-FL Env-pseudotyped viruses produced in primary PHA/IL-2-activated CD4TL from a healthy donor have a greater capacity to infect primary CD4TL than MDMs. Two independent experiments were carried out on the cells from two distinct donors. In each experiment, CD4TL and MDMs were obtained from the same individual. Experiments on both cell types were carried out using the same virus load (30 ng of p24). Means ± SEM are shown. **D)** The supernatant of producer Jurkat cells does not influence infectivity of viruses. Similar infectious doses (100,000 RLU) of supernatants from producer HEK 293T or Jurkat cells, or JK-derived viruses centrifuged and resuspended in fresh medium were exposed on MDMs. Infectivity measured 48 h post-inoculation is expressed relative to infectivity of HEK-derived viruses. Means ± SEM are shown. *, $P < .05$; **, $P < .01$ using the Mann-Whitney U-test.

pseudotyped viruses (**S4 Fig**). As depicted in **Fig 4A**, we observed that some viruses produced in Jurkat cells, pseudotyped with R5-25, -34 or JR-CSF Env, were more efficient to infect primary T cells compared to those produced in HEK 293T cells, while the 5 other viruses displayed similar infectivity regardless of the cell-type of production. Then, as in **Fig 1**, we used 35,000 RLU of viruses produced either in HEK 293T cell or in Jurkat cells to inoculate MDMs. As shown in **Fig 4B**, except JR-FL Env-pseudotyped viruses, none of the tested viruses produced in Jurkat cells were significantly able to infect MDMs. Surprisingly, viruses pseudotyped with Env R5-13 or Env R5-34, which were consistently described as M-tropic when produced in HEK 293T cells (**Fig 1** and reference [31]), failed to infect MDMs when derived from Jurkat cells, regardless of the blood donor. Even viruses pseudotyped with the prototypic M-tropic Env JR-FL showed a lower infectivity in MDMs when produced by Jurkat cells (**Fig 4B**). Similarly, we observed that cell-free JR-FL Env-pseudotyped viruses produced by primary CD4TL efficiently infected primary CD4TL used as targets but only modestly MDMs (**Fig 4C**).

In these experiments, as well as for results reported in **Fig 1**, the supernatants from the producer cells underwent gentle centrifugation to remove cellular debris and were then directly added to the target cells (either primary CD4+ T cells or MDMs). We therefore asked if some factors present in the supernatant of infected Jurkat cells might be responsible for the reduced ability of the viruses produced by these cells to infect MDMs. To investigate this, we compared the capacity of the JR-FL Env-pseudotyped virus particles produced in Jurkat cells to infect MDMs before or after clearance of the cell-culture supernatant by ultracentrifugation and resuspension of the viruses in fresh culture medium. Similar infectious quantities of these virus preparations (100,000 RLU in CD4TL) were used to inoculate MDMs. As shown in **Fig 4D**, removing the supernatant of Jurkat producer cells did not restore the capacity of the JR-FL Env-pseudotyped viruses to infect MDMs. Collectivity, these results suggest that intrinsic properties of CD4+ T cell-derived viruses make them less efficient at infecting MDMs compared to viruses produced by HEK 293T cells. These results explain our observations above showing that MDMs are not infected when exposed to the cell-culture supernatants of infected T cells (see **Figs 2** and **3**), even for high virus load, as measured by p24 (our results here and references [35,47]). This also strongly suggests that efficient transfer of HIV-1 from T cells to MDMs is not due to enhanced tropism of T cell-derived Envs for MΦ.

## MDM infection through virus cell-to-cell transfer from infected CD4+ T cells is linked to increased efficiency of CD4 and CCR5 usage

Our results above indicated that constraints that limit cell-free viruses in their ability to infect MDMs were overcome when viruses were cell-to-cell transferred to MDMs from infected T cells. The majority of the cell-free viruses studied here were unable to infect macrophages due to a blockage at the viral entry step (see **Fig 1**). This is consistent with previous studies, including ours, showing that macrophage-tropism is mostly related to the increased ability of Envs to use entry receptors [22–24,26,28,29,31,61]. We therefore addressed the question whether improved efficiency of MDM infection through virus cell-to-cell transfer was due to an increased efficiency of CD4 and/or coreceptor usage. For that purpose, we compared the dependence of HIV-1 Envs on receptor expression levels on MDMs in cell-free or cell-to-cell viral transfer from infected Jurkat cells (**Figs 5 and 6**). We reasoned that more efficient use of receptors by these Envs should result in their decreased dependence on receptor expression levels.

We first pretreated MDM targets with increasing concentrations of the CCR5 entry inhibitor MVC to gradually decrease the availability of the coreceptor at the cell surface, before exposure to cell-free viruses pseudotyped with the M-tropic Env JR-FL or to Jurkat cells infected with the same virus (**Fig 5A**). Infection of MDMs was then analyzed by flow cytometry after staining for intracellular Gag p24, as above. Of note, the titers of cell-free viruses and the degree of infection of Jurkat cells were adjusted to obtained similar proportions of infected MDMs (10% of Gag+ MDMs, **Fig 5B**) in control conditions (*i.e.* in the absence of MVC). Under these conditions, infection by cell-to-cell virus transfer tended to be more resistant to inhibition by MVC than infection with cell-free viruses, but this difference did not reach statistical significance (**Fig 5C**). With the exception of one experiment out of 4 (red symbols in **Fig 5C**), the inhibitor prevented MDM infection with an $IC_{50}$ in the low nanomolar range, regardless of the mode of infection ($IC_{50}$ = 4.1 ± 2.9 and 1.3 ± 0.6 nM for the CF and CTC conditions, respectively). This range was close to the MVC affinity constant for CCR5 and its antiviral potencies measured under other experimental conditions [62–64].

The fact that the two modes of infection of MDMs were similarly sensitive to MVC suggested that they do not drastically alter the way viruses use CCR5. However, we sought to

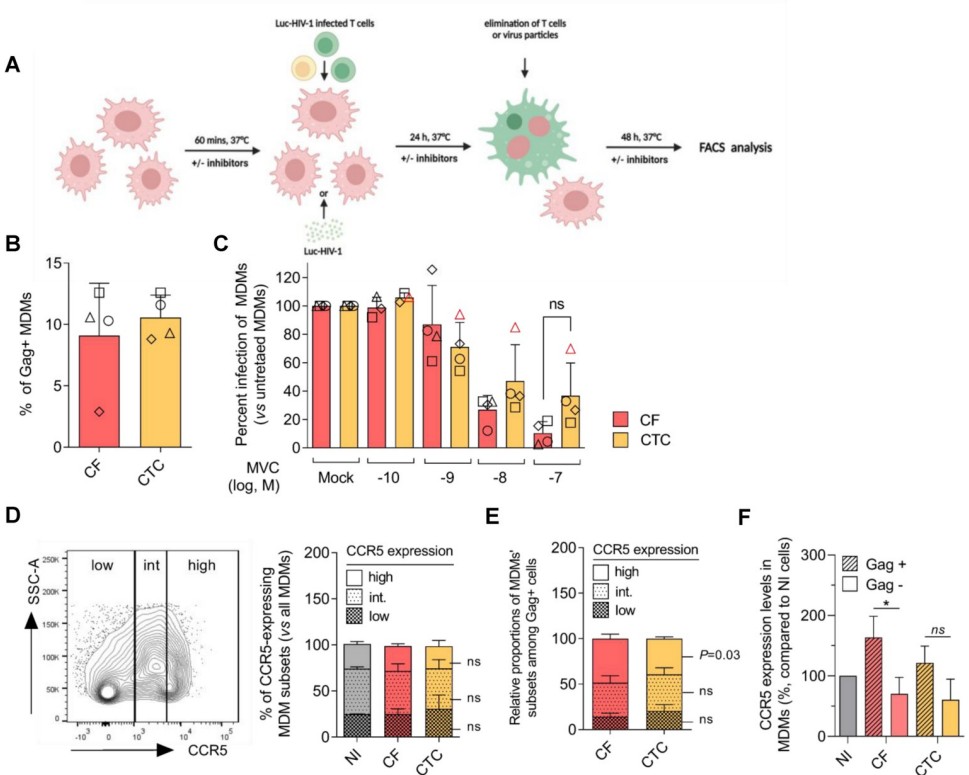

**Fig 5. Increased efficiency of CCR5 use in HIV-1 transfer between infected T cells and MDMs. A)** Experimental scheme of experiments. **B)** The experiments analyzing CCR5 dependence were performed with similar proportion of infected MDMs (Gag+ MDMs) exposed to cell-free JR-FL Env-pseudotyped viruses or Jurkat cells infected with the same viruses. **C)** Dose-dependent inhibition by maraviroc (MVC) of MDM infection through cell-to-cell viral transfer (CTC) or by cell-free viral particles (CF). MDMs were incubated at 37˚C for 60 min with the indicated doses of MVC, then exposed for 24 h to infected Jurkat cells or viral particles (in the presence of the same doses of MVC), washed and incubated for further 48 h à 37˚C (still with MVC). Infectivity was then determined by flow cytometry by quantifying the amount of Gag+ MDMs. Results represent means ± SD of 4 independent experiments, expressed as percent infectivity relative to infectivity determined in the absence of MVC (100%). **D)** The left panel represents the gating strategy of MDMs expressing low, intermediate or high levels of CCR5, after CTC or CF infection in the absence of MVC, as detailed above. A representative flow cytometry plot of cell side scatter as a function of CCR5 expression level revealed with a PE-Vio 770-conjugated anti-CCR5 mAb is shown. Data were acquired using a BD LSRII flow cytometer. Right panel: Relative proportions of MDM subsets expressing low, intermediate or high levels of CCR5 among all MDMs exposed, or not (NI), to infected Jurkat cells (CTC) or to cell-free viruses (CF). **E)** Relative proportion of MDMs expressing low, intermediate or high levels of CCR5 among total Gag+ MDMs after cell-to-cell viral transfer (CTC) or cell-free (CF) infection. Results are expressed as percent of total Gag+ cells. **F)** CCR5 expression levels at the surface of Gag+ or Gag- MDMs after CTC or CF infection, compared to untreated MDMs (NI). Receptor expression levels were determined by FACS analysis, measuring the geometric mean fluorescence intensity of PE-Vio770-conjugated anti-human CCR5 bound to the different MDM populations. Statistics: Mann-Whitney U-test: *, $P < 0.05$; ns: not significant.

analyze the question of the efficiency of CCR5 usage more directly. To this end, we analyzed the proportion of Gag+ MDMs (*i.e.* infected cells) as a function of CCR5 cell-surface expression level in infection experiments using cell-free viruses or coculture with infected Jurkat cells. We selected MDM subsets expressing low, intermediate or high levels of CCR5 (**Fig 5D**, left panel) and then compared their permissiveness to infection in the two experimental conditions (**Fig 5E**). In both cases, we observed an enrichment of Gag+ MDMs among high-CCR5-expressing cells (compare right panel of **Fig 5D** with **Fig 5E**). This is consistent with the view that infection of MDMs is favored when CCR5 expression increases. Interestingly, this

was less marked in the case of MDM infection by virus cell-to-cell transfer, compared to infection with cell-free viruses (**Fig 5E**). Indeed, the proportion of Gag+ MDMs that express high levels of CCR5 was statistically lower in the cell-to-cell conditions than in the cell free conditions (39.2 ± 2.1 *vs* 48.5 ± 4.9%, *P* = 0.03). Conversely, the proportions of Gag+ cells among intermediate- and low-CCR5-expressing MDMs tended to be increased in the cell-to-cell condition compared to cell free, albeit these differences were not statistically significant. Consequently, MDMs infected with cell-free viruses expressed more CCR5 than MDMs infected through virus cell-to-cell transfer (**Fig 5F**). Taken together, these data suggest that infection of MDMs through virus cell-to-cell transfer is less dependent on CCR5 expression level than infection with cell-free viruses, suggesting greater efficiency in CCR5 usage. However, this increase in the efficiency of CCR5 use does not seem to be important enough to significantly decrease sensitivity of viruses to MVC.

We next investigated whether virus cell-to-cell transfer to MDMs was linked to increased efficiency of CD4 usage. To this end, MDMs were pretreated with varying concentrations of the neutralizing anti-CD4 mAb Q4120, and then infected by cell-free JR-FL Env-pseudotyped viruses or by coculture with Jurkat cells infected with viruses pseudotyped with JR-FL Env or JR-CSF Env. Indeed, we and others previously showed that cell-free non-M-tropic viruses (e.g. JR-CSF) were more sensitive to inhibition by Q4120 than M-tropic viruses (JR-FL) [25,31]. Here, we hypothesized that if cell-to-cell virus transfer to MDMs proceeds with increased efficiency of CD4 use, compared to infection by cell-free viruses, this might translate into lower sensitivity to Q4120. As above, the inhibition experiments with Q4120 were carried out under conditions of similar levels of MDM infection, regardless of the mode of infection and the viruses used (**Fig 6A**). Infection of MDMs through coculture with infected Jurkat cells tended overall to be more resistant to inhibition by Q4120 than infection by cell-free viruses (**Fig 6B**). This was observed with Jurkat cells infected with JR-FL or JR-CSF Env viruses. Non-linear regression of individual inhibition curves gave increased $IC_{50}$ values for Q4120 in the cell-to-cell conditions, compared to the cell-free conditions (mean $IC_{50}s$ = 0.27, 1.7 and 1.18 nM for the CF(FL), CTC(FL) and CTC(CSF) conditions, respectively). However, results were highly variable between MDM preparations and these differences only reached statistical significance at the highest Q4120 concentration tested (1 nM).

We then asked whether the mode of infection or the Env-pseudotyped viruses were affected by the level of CD4 expression on MDMs. Of note, we observed that CD4 expression level was down-modulated on MDMs at 72 h post-exposure to infected Jurkat cells or cell-free viruses (**Fig 6C**), in contrast to CCR5 expression (see **Fig 5**). This is consistent with previous works showing that CD4 was down-modulated in HIV-1-infected cells [65,66]. Accordingly, the proportion of CD4-positive cells was decreased among infected (Gag+) cells, compared to uninfected cells (**Fig 6D**). This effect was of similar magnitude regardless the mode of infection and the nature of Env. However, the down-modulation of CD4 after MDM infection did not change the relative proportions of low, intermediate and high CD4-expressing MDMs (**Fig 6E**). Under these conditions, as for CCR5 use, cell-free viruses pseudotyped with JR-FL Env preferentially infected MDMs that express high levels of CD4, as revealed by the enrichment of Gag+ MDMs in high CD4-expressing cells (**Fig 6F**). This effect was however less marked when infection proceeded through virus cell-to-cell transfer where the proportion of MDMs expressing low and moderate levels of CD4 was predominant when using JR-FL Env and, somewhat unexpectedly, even more with viruses pseudotyped with the non-M tropic Env JR-CSF. These data are consistent with the view that transfer of HIV-1 between infected T cells and MDMs proceeds through increased efficiency of CD4 use, compared to infection by cell-free viruses.

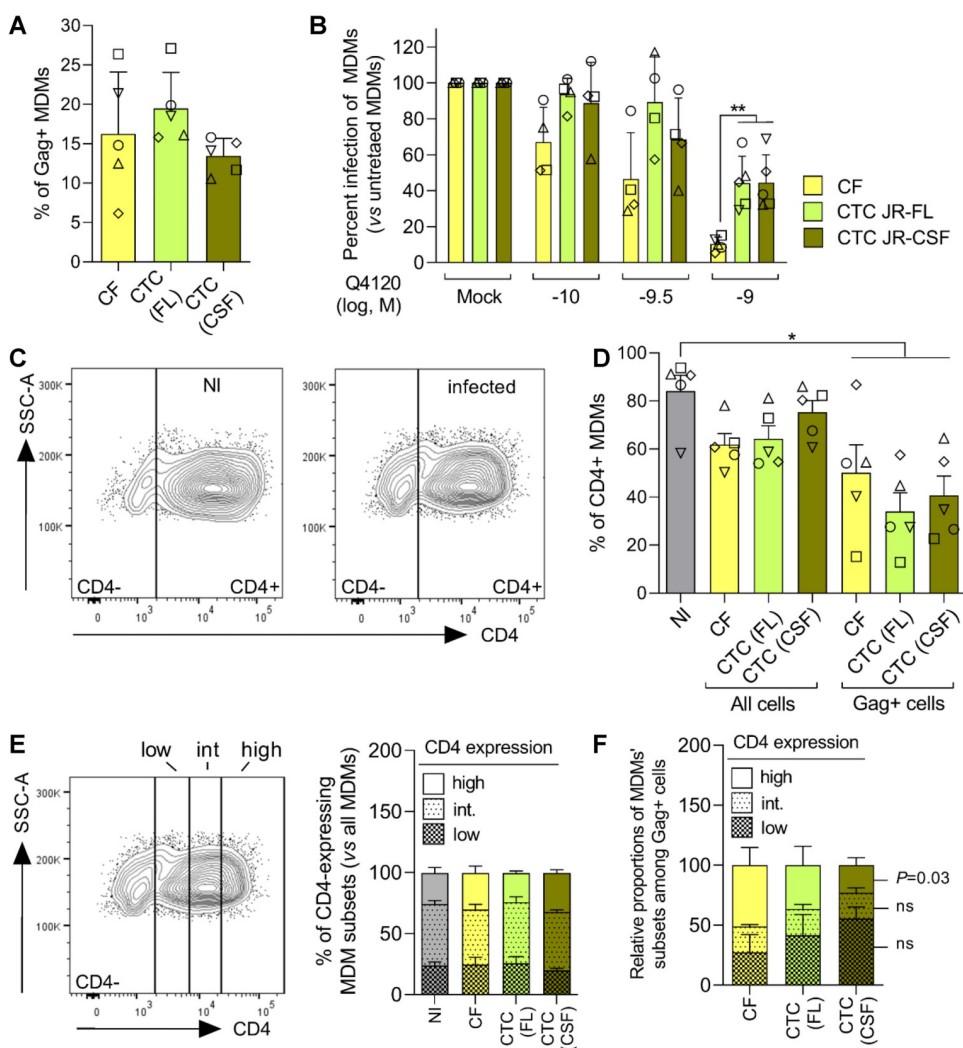

**Fig 6. Increased efficiency of CD4 use in HIV-1 transfer between infected T cells and MDMs. A)** The experiments analyzing CD4 dependence were performed with similar proportion of Gag+ MDMs exposed to cell-free (CF) JR-FL Env-pseudotyped viruses or Jurkat cells infected with JR-FL (FL) or JR-CSF (CSF) Env-pseudotyped viruses (CTC). **B)** Dose-dependent inhibition by Q4120 of CTC or CF MDM infection, carried out as described in the experimental scheme of **Fig 5A**. Error bars: SD. **C)** Representative flow cytometry plots of cell side scatter as a function of CD4 expression level at the surface of untreated (NI) or virus-exposed (infected) MDMs. **D)** Proportions of CD4+ cells within untreated MDMs (NI), or MDMs exposed to infected Jurkat cells (CTC) or cell-free viruses (CF). In the latter case, results were analyzed for all MDMs (Gag- and Gag+ cells) or among Gag+ MDMs only. Results are means ± SEM of 5 independent experiments. **E)** The left panel represents the gating strategy of MDMs expressing low, intermediate or high levels of CD4, after cell-to-cell (CTC) or cell-free (CF) infection in the absence of inhibitor, as detailed above. CD4 expression level was determined using a BV786-conjugated anti-CD4 mAb on a BD Fortessa flow cytometer. Right panel: Relative proportions of MDM subsets expressing low, intermediate or high levels of CD4 among all MDMs exposed, or not (NI), to infected Jurkat cells (CTC) or to cell-free viruses (CF). **F)** Relative proportions of MDMs expressing low, intermediate or high levels of CD4 among total Gag+ MDMs after CTC or CF infection. Statistics: Mann-Whitney U-test; *, $P < .05$; ns: not significant.

## Transmitted/founder viruses can be efficiently transferred from T cells to macrophages through cell-cell fusion

Finally, because R5 T/F viruses have often been characterized as non-M/T-tropic in cell-free infection assays, we investigated whether they could be transferred from infected T cells to MΦ through cell-cell fusion. As previously, Jurkat cells were infected with infectious molecular

clones expressing 4 different T/F Envs (CHO58, THRO, REJO and RHPA, described in ref. [19]). Infected cells were cocultured for 6 or 24 h with MDMs and virus cell-to-cell transfer was quantified by flow cytometry after intracellular Gag staining. As controls, cell-culture supernatants of infected Jurkat cells were used to infect MDMs for comparison with cell-free virus infection. As shown in **Fig 7A and 7B**, the 4 T/F viruses were significantly transferred to MDMs when cocultured with infected Jurkat cells, with the highest level of virus transfer observed with THRO and REJO. Simirlarly, the 4 T/F viruses were efficiently transferred to MDMs when purified blood primary CD4+ T cells were used as virus-donor T cells (**Fig 7C**). In contrast, no or low level of virus transfer was detected when MDMs were infected with cell-free viruses (**Fig 7A and 7C**, red bars), confirming that virus cell-to-cell transfer is more efficient than cell-free infection.

To visualize whether T/F viruses were transferred to MDMs by cell-cell fusion with virus-donor T cells, infected Jurkat or primary CD4+ T cells were preloaded with CellTracker and cocultured for 6 h with MDMs before intracellular Gag and DRAQ5 staining for fluorescence microscopy analysis. Interestingly, MDMs cocultured with Jurkat or primary CD4+ T cells infected with THRO or REJO exhibited a strong and diffuse cytoplasmic Gag staining (**Figs 7D and S5A**) and almost all contained at least 2 nuclei (**Figs 7E and 7G, and S5B and S5C**), including at least 1 CellTracker+ nucleus (**Fig 7F and 7H**). The formation of MGCs was inhibited when cocultures were run in the presence of MVC or T20, but not in the presence of AMD3100 (**Fig 7I**), indicating that cell-cell fusion mediated by these two T/F viruses depends on Env/CCR5 interactions and Env-mediated fusion between infected T cells and MDMs. In contrast, most of the target MDMs cocultured with Jurkat or CD4TL cells infected with CHO58 or RHPA viruses showed a cytoplasmic dotted Gag staining (**Figs 7D and S5A**) and were mononucleated (**Figs 7E and 7G and S5B and S5C**). This suggests that these two T/F viruses can be captured by MDMs and accumulate in cytoplasmic compartments after cell-to-cell contacts with infected T cells. This finding is in accordance to flow cytometry data showing that the CHO58 and RHPA viruses can be transferred (see Fig **7A–7C**), but by a process that does not depend on cell-cell fusion.

Altogether, these findings indicate that some T/F viruses, even those classified as non-M-tropic by cell-free infection assays, can be transferred to MΦ from infected T cells through mechanisms depending or not on cell-cell fusion. The fusion between infected T cells and MΦ, however, leads to a more massive spread of viruses.

## Discussion

Macrophages are increasingly recognized as HIV-1 cellular targets involved in virus dissemination and formation of viral tissue reservoirs [1–8,13,32,33]. It is therefore paradoxical that most HIV-1 isolates do not or only inefficiently infect MΦ in cell-free virus infection assays of MDMs or surrogate cells [16,67]. However, our results indicate that these assays, commonly used to discriminate M-tropic and non-M-tropic viruses, fail to recapitulate all the mechanisms by which HIV-1 isolates can infect MΦ. We show that a large panel of CCR5- and/or CXCR4-using viruses acting as non-M-tropic viruses in cell-free virus infection assays (**Fig 1** and refs [31,56]) can productively infect MΦ when transferred from infected CD4+ T cells, similarly as *bona fide* M-tropic viruses. These results corroborate previous results showing that many viruses isolated from MDMs of patients' PBMCs have a T-cell origin, and as cell-free particles can successfully infect CD4+ T lymphocytes but not MΦ [68].

Previous works showed that phagocytosis or engulfment of infected T cells by MDMs may contribute to productive infection of MDMs by weakly M-tropic viruses, including some T/F viruses [42,43]. The T cell/MDM viral transfer we document here through a two-step cell-cell fusion process leading to formation of infected MGCs differs from the previous mechanism in

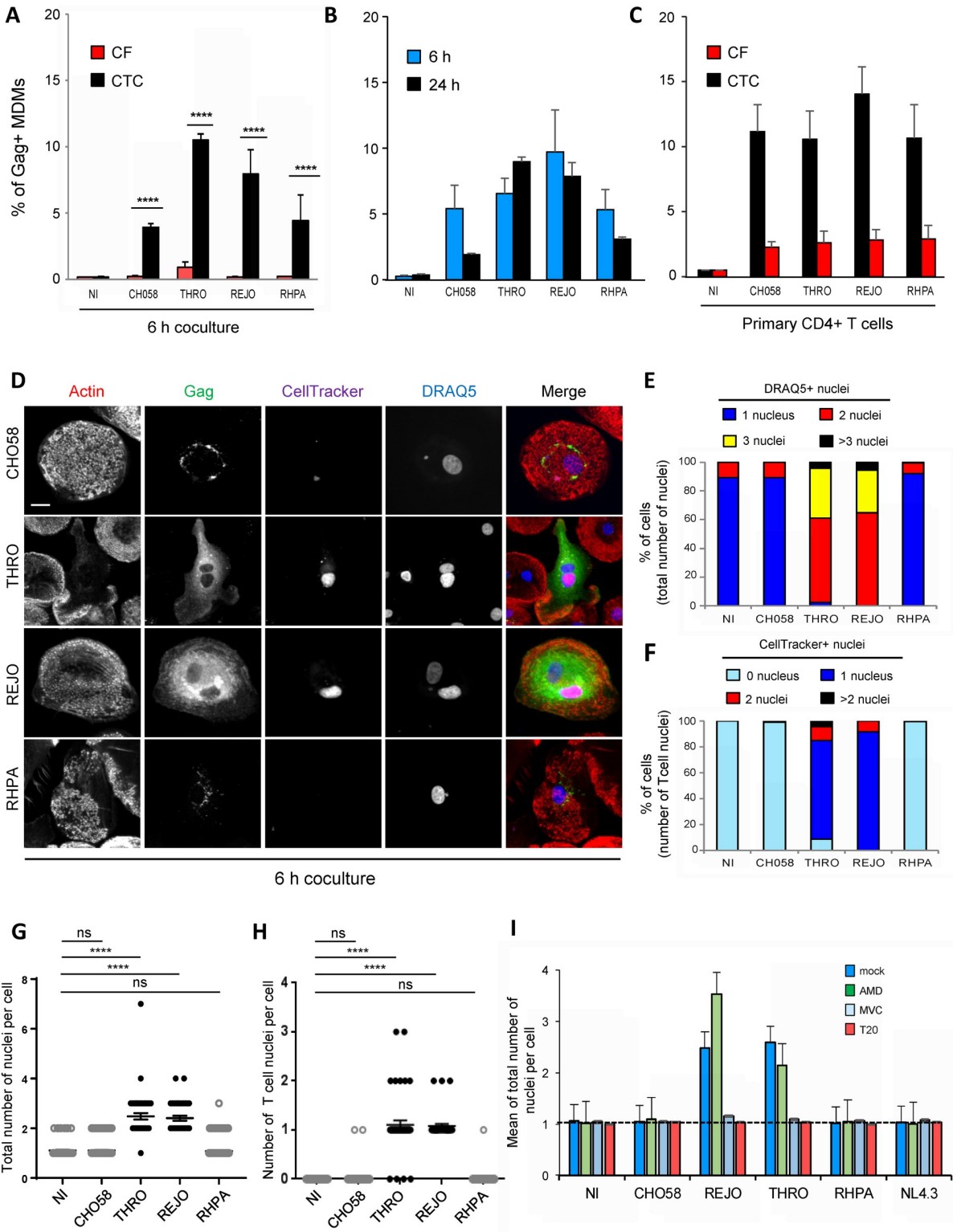

**Fig 7. Cell-to-cell transfer to macrophages of CCR5-using transmitted/founder HIV-1 infectious molecular clones. A-C)** MDMs were exposed to cell-free T/F virus-containing supernatants produced from infected Jurkat cells or cocultured with the corresponding infected Jurkat (**A** and **B**) or primary CD4+ T (**C**) cells for 6 or 24 h. The percentage of Gag+ MDMs was determined by flow cytometry. Non-infected MDMs or MDMs cocultured with non-infected Jurkat or primary T cells were used as negative controls (NI) of cell-free or cell-to-cell infection, respectively. In **A)**, the results show the percentage of Gag+ MDMs after 6 h of cell-free (CF, red bars) or cell-to-cell (CTC,

black bars) infection. In **B**), the results show the percentage of Gag+ MDMs after coculture of infected Jurkat cells for 6 (blue bars) or 24 h (black bars) with target MDMs. In **C**) the results show the percentage of Gag+ MDMs after 24 h of incubation with cell-free virus-containing supernatants (corresponding to 500 ng of p24, red bars) or cocultured with infected primary CD4+ T cells (black bars). **D-I**) Jurkat cells infected with the indicated virus clones and prelabeled with CellTracker were cocultured for 6 h (**D-H**) or 24 h (**I**) with MDMs. After elimination of Jurkat cells, MDMs were stained with anti-Gag, phalloidin (actin) and DRAQ5 (nuclei), and analyzed by confocal microscopy (scale bar, 10 μm). Representative images are shown in (**D**). The number of T cell nuclei (CellTracker+) as well as the total number of nuclei (DRAQ5+) per Gag+ MDM were quantified from images on at least 50 cells. MDMs cocultured with non-infected Jurkat cells were used as negative controls (NI). In **E**), results are expressed as the percentages of Gag+ MDMs with 1, 2, 3 or more than 3 DRAQ5(+) nuclei. In **F**), results are expressed as the percentages of Gag+ MDMs with 0, 1, 2, or more than 2 CellTracker(+) nuclei. In **G**), results are expressed as the number of DRAQ5(+) nuclei per MDM cocultured with infected Jurkat cells. In **H**), results are expressed as the number of CellTracker(+) nuclei per Gag+ MDM cocultured with infected Jurkat cells; each dot corresponds to 1 cell. In **I**), results are expressed as the means of total nucleus number (DRAQ5+) per Gag+ MDM pretreated or not (dark blue bars, mock) with AMD3100 (green, AMD), maraviroc (light blue, MVC), or T20 (red), and cocultured with infected Jurkat cells. In **A**, **B**, and **I**), the results represent means of at least 4 independent experiments performed with MDMs of at least 4 different donors, while in **C**) the results are the means of 2 independent experiments performed with MDMs of 2 different donors; error bars represent 1 SEM. In **D-H**), the results shown are representative of at least 4 independent experiments performed with MDMs from at least 4 different donors; horizontal bars in (**G**) and (**H**) represent means +/- 1 SEM. Statistical significance was determined using the Mann-Whitney U-test (ns, not significant, $P > 0.05$; ****, $P < 0.0001$).

several ways. In contrast to what we observe, phagocytic uptake of infected T cells does not appear to allow infection of MDMs by X4 T-tropic viruses [44]. Phagocytosis also more commonly involves dead or dying virus-donor infected T cells and occurs through a mechanism independent of the interactions between Env and cellular receptors (i.e., CD4 and CCR5/ CXCR4), even though the subsequent step leading to productive infection and spreading in MDM targets requires Env [42]. Extending our previous works in which laboratory-adapted, R5 M-tropic viral strains were used [45,47], we show here that primary HIV-1 strains isolated from infected patients, regardless of their viral tropism (R5 and/or X4), can be effectively transferred from infected T cells to MΦ through Env-dependent cell-cell fusion. Then, the newly formed T cell/MΦ fused cells fuse with surrounding uninfected MΦ, leading to formation of MGCs and massive dissemination of the virus. Both sequential cell-cell fusion processes are dependent on interactions of Env expressed on virus-donor T cells with CD4 and the chemokine receptors expressed on MΦ, since they are inhibited by anti-Env antibodies, as well as by drugs such as the fusion inhibitor T20 targeting the transmembrane gp41 subunit, but also by the specific CCR5 and CXCR4 antagonists MVC and AMD3100 (this study, and Refs [45,47]). In addition, the viral tropism of the viral strains analyzed here, defined by the coreceptor usage, is usually maintained between cell-free and cell-to-cell infection processes. Our data indicate that infected T cell-to-MDM viral transfer is less sensitive to inhibition by the anti-CD4 mAb Q4120, and to a lesser extent by the CCR5 antagonist MVC. This provides evidence indicating that Envs expressed on infected T cells interact more efficiently with entry receptors expressed on MΦ than cell-free virus-associated Envs. In addition, cell-to-cell transfer of R5 viruses, including viruses expressing the prototypic non-M-tropic JR-CSF Env, is less dependent on CD4 and CCR5 cell-surface expression levels on MDMs than infection with cell-free viruses. Enhanced Env/receptor interactions may help increase the capacity of HIV-1 Envs, and particularly non-M-tropic Envs, to trigger cell-cell fusion of infected T cells with MΦ. Consistently, cell-free particles of most non-M-tropic viruses inefficiently use entry receptors and are blocked in their capacity to infect MΦ at the viral entry stage, in agreement with the results of our virus-cell fusion assays (see **Fig 1**). Conversely, M-tropic viruses often showed reduced dependence on CD4 and/or coreceptors for viral entry, and increased resistance to inhibitors targeting either of these receptors [22–31,69]. Lower dependence on virus entry receptors may also be responsible for the capacity of the X4 strains defined as T-tropic in cell-free infection assay to be efficiently transferred from infected T cells to MDMs through cell-cell fusion, while still using CXCR4 as coreceptor. Interestingly, our data also suggest that this increase in the efficiency of entry receptor usage may occasionally be accompanied by

changes in viral tropism. We observed that cell-to-cell transfer of viruses pseudotyped with X4-1 Env was inhibited by a mixture of AMD3100 and MVC, but not by AMD3100 or MVC alone, indicating that X4-1 Env could use CXCR4 and CCR5 to mediate cell-cell fusion between infected T cells and MDMs. This contrasts with our previous phenotypic determination of X4-1 Env viral tropism based on cell-free virus infection of U87-CD4 reporter cells expressing CCR5 or CXCR4 that indicated exclusive usage of CXCR4 by this Env [56]. Additionally, we reported that AMD3100 or CXCL12 (the CXCR4 natural ligand) fully inhibited infection of primary CD4+ T cells by cell-free X4-1 Env-pseudotyped viruses ([56], and here). These intriguing results could reflect contextual usage of coreceptors by X4-1 Env depending on the nature of the target cells (e.g. U87 and CD4TL *vs* MDMs) and/or changes in the properties of Env/coreceptor interactions in cell-to-cell virus transfer compared to cell-free infection.

Several mechanisms could lead to enhancement of Env/receptor interactions within infected T cell/MDM conjugates. Strikingly, our observations are similar to previous results showing that cell-to-cell transfer of HIV-1 between CD4+ T cells across the virological synapse is less sensitive to anti-CD4 Q4120 antibody than in cell-free infection [70]. Multiple mechanisms have been proposed to explain the higher efficiency of infection of T cells through the virological synapse compared to cell-free infection [37,39,70], and could also be involved in virus cell-to-cell transfer from infected T cells to MDMs. In particular, formation of the virological synapse is characterized by a redistribution and clustering of HIV-1 receptors and Envs at the surface of the target and virus-donor cells, respectively [71,72]. Although similar processes in the context of infected T cell-MDM contacts remain to be demonstrated, they could similarly contribute to reinforce Env/receptor interactions and decrease the dependence on receptor expression levels. Indeed, clustering of receptors may increase their surface density at the surface of MDMs, which has been reported to strengthen Env binding and to result in more rapid Env-mediated fusion kinetics and increased resistance to virus entry inhibitors [73–75]. In addition, redistribution of coreceptors into distinct membrane environments might change their conformational states, thereby influencing Env binding and virus entry, as we recently illustrated [31]. Finally, clustering of Envs at the surface of infected T cells could also facilitate virus transfer by the cell-cell fusion process. While non-M-tropic Env trimers adopt a conformation distinct from that of M-tropic Envs with a weaker affinity for CD4 [61,76,77], a local increase of Env molecules at the T cell/MDM contact site could overcome this limitation and favour cell-cell fusion. In contrast, cell-free virus particles, known to express limited amounts of Env trimers at their surface, could not overcome this limitation [78,79].

HIV-1 cellular tropism, including MΦ-tropism, has often been studied using virus particles produced in HEK 293T cells [22,26,61,80]. Interestingly, our results show that viruses produced in T cells are less able to infect MΦ, while both types of viruses have similar infectivity in primary CD4TL. The reasons for this observation remain unclear, but changes in viral membrane composition, which can influence infectivity by regulating Env conformation [81,82], may explain the decreased infectivity in MΦ of T cell-produced viruses compared to those produced from HEK 293T cells. Nevertheless, these results suggest that in anatomical sites of virus spread, such as lymphoid tissues, where infected CD4+ T cells coexist with MΦ, infection of MΦ more likely occurs by virus cell-to-cell transfer rather than by cell-free viruses produced in the extracellular medium by infected T cells.

Retrospectively, several results of the literature suggested that cell-to-cell transfer of HIV-1 from infected T cells to MΦ might play a crucial role in different aspects of the pathophysiology of HIV-1 infection. Initially, however, it was assumed that M-tropism might emerge as a consequence of CD4 T cell paucity. Actually, works using humanized myeloid-only mice showed that MΦ can sustain systemic replication of HIV-1 and formation of viral reservoirs in the absence of T cells [13,33]. However, HIV-1 infection in this model could be achieved with

M-tropic viruses but not with non-M-tropic viruses. Earlier, the role of MΦ as genuine virus-producing cells was shown in rhesus macaques infected with a highly pathogenic SIV/HIV chimera (SHIV) [3]. In these monkeys, high levels of virus replication persisted in MΦ from different tissues (lymph nodes, gastrointestinal tract, spleen, liver, kidney) after initial rapid loss of CD4+ T cells, observed within the first 3 weeks. Subsequent works in macaques in which CD4+ T cells have been experimentally depleted using anti-CD4 antibodies suggested that viral replication in myeloid cells was favored in the absence of T cells [83,84], likely as a result of a cellular tropism shift of viral Envs from T cells to MΦ [84, 85]. These results explain why M-tropic viruses are often compartmentalized in tissues with low CD4+ T cell content such as the brain [27,58]. Based on these observations, Calantone and coll. [32] hypothesized, still using SIV-infected monkeys, that infection of MΦ and related myeloid cells might be favored in tissues where CD4+ T cells are in low amounts, as compared to tissues with abundant T cells such as lymphoid tissues. Surprisingly, the opposite result was observed, and viral DNA in myeloid cells was almost exclusively found in tissues where CD4+ T cells were abundant such as the spleen and mesenteric lymph nodes. These authors also found that infected myeloid cells contained T cell markers, consistent with infection of these myeloid cells by cell-cell fusion with and/or engulfment of infected T cells [32]. Interestingly, it was also found that *in vivo* infection of myeloid cells in infected animals occurred irrespective of expression of the SIV Vpx auxiliary protein known to counteract the antiviral host cell restriction factor SAMHD1 highly expressed in myeloid cells [86–89]. This is reminiscent of our previous work showing that HIV-1 spread in MΦ or dendritic cells through initial cell-cell fusion with infected T cells bypasses the post-entry SAMHD1 restriction for efficient virus replication in these myeloid cells [47]. Since cell-cell fusion infection overcomes both entry and post-entry blocks for most R5 and X4 tropic strains, including those defined as R5 or X4 T-tropic by cell-free assays, it may be a prevalent mode for HIV-1 spreading *in vivo* in MΦ and related myeloid cells in tissues rich in CD4+ T cells. In agreement, a recent report shows that infected CD4+ T cells promote infection of alveolar macrophages by non-M-tropic primary HIV-1 isolates [7].

Other circumstances in which infection of MΦ through viral transfer from infected T cells could be considered. Indeed, previous results showed that T-tropic viruses are often present in the CNS together with compartmentalized M-tropic viruses [27,90]. Brain T-tropic viruses are thought to originate from translocation of infected blood CD4+ T cells, an early process after HIV-1 transmission, preceding virus compartmentalization in the brain and exacerbated in conditions of CNS inflammation [91]. It was proposed that viruses continuously released from translocated T cells contribute to initial infection of brain MΦ and microglial cells [16]. To this regard, we and others previously described that some T-tropic viruses from the blood can infect MΦ to some extent, in particular through usage of particular CCR5 conformations [29,31], and might therefore contribute to establishment of infection in the CNS. Present results suggest that cell-cell fusion of brain myeloid cells with translocated infected T cells could also contribute to this process, at high efficiency and for a wide range of viruses. Our data also suggest that infection of MΦ (and/or dendritic cells) from infected T cells might play a role in HIV-1 transmission. As mentioned above, cell-free virus infection assays led to the conclusion that sexually-transmitted R5 T/F viruses have little or no ability to infect MΦ, suggesting that MΦ do not contribute to virus early dissemination after transmission [18,19,92,93]. However, HIV-1 establishes new infections at mucosal surfaces and then in neighboring draining lymph nodes that are both heavily populated with CD4+ T cells and myeloid cells including MΦ. Several works have also reported that MΦ can be infected at sites of HIV-1 transmission [8,94–96], raising the question of how they became infected. Our results support the view that, at the site of transmission, resident MΦ (and/or DCs) may be efficiently infected by T/F viruses through cell-cell fusion with infected T cells (see **S5D** and

S5E Fig) and then contribute to virus dissemination. While R5 viruses, including T/F viruses, predominate in early, acute and asymptomatic phases of infection, dual-tropic R5/X4 and X4 variants can emerge at later stage of infection in some patients, when the proportion of CCR5-positive, memory CD4+ T cell drops in the peripheral blood, but resting naïve CD4+ T cells expressing mainly CXCR4 are still maintained (For recent reviews see references [16,97]). Since we show that R5/X4 and X4 viruses are very efficiently transferred to MΦ, we can propose that infected naïve CD4+ T cells could then transfer CXCR4-using viruses to tissue MΦ, thereby contributing to dissemination at later stage.

In conclusion, our findings show that MΦ-tropism of primary HIV-1 isolates, currently based on cell-free infection assays, needs to be revisited to reflect the capacity of both CCR5- and CXCR4-using non-M-tropic variants, including some T/F viruses, to spread efficiently in myeloid cells via cell-to-cell transfer from infected T cells. While the viral tropism of viral strains, defined by coreceptor usage, is mostly maintained in cell-free and cell-to-cell infection, the initial block of virus entry sustaining efficient replication in MΦ is overcome via cell-to-cell viral transfer from infected T cells. As discussed, these findings likely impact different stages of the pathophysiology of HIV-1 infection regarding i) viral sexual transmission through genital and rectal mucosa, ii) viral spreading in target cells and in lymphoid and non-lymphoid tissues, and iii) establishment of the viral tissue reservoirs.

## Material and methods

### Ethics statement

Blood samples from anonymous healthy donors were purchased at *"Etablissement Français du Sang"* (EFS, the French National Blood Agency). Donors provided written informed consent to EFS at the time of blood collection. The primary R5 and X4 Envs used for virus phenotyping were isolated from biological virus clones of PBMCs collected in individuals of the Amsterdam Cohort Studies (ACS) on HIV-1 and AIDS (a gift from Pr H. Schuitemaker, University of Amsterdam, the Netherlands). Samples were used in accordance with legal and ethical conditions we previously detailed [31,56]. These Envs and the patients from whom they were isolated were described in our previous works [31,56].

### Plasmids and reagents

T/F HIV-1 infectious molecular clones CH058, THRO, REJO, and RHPA have been already described [19], and were obtained from the AIDS Research and Reference Reagent Program, Division of AIDS, NIAID. For construction of pseudotyped replicative viruses, HIV-1 Envs were cloned in the pNL-SacII-lacZ/env-Ren proviral vector, as described previously [31]. The following antibodies were used: RD1- or fluorescein isothiocyanate (FITC)-conjugated anti-Gag (clone KC57; Beckman Coulter), Alexa Fluor 647- or Fluor 555-conjugated phalloidin (Life Technologies), PE-Vio770-conjugated anti-human CCR5 (clone REA245, Miltenyi Biotec), PE-conjugated anti-human CD11b (clone REA713, Miltenyi Biotec), BV786-conjugated anti-human CD4 (clone SK3, BD Biosciences). The anti-CD4 neutralizing mAb Q4120 was provided by Dr. Q. Sattentau and the NIBSC Centralised Facility for AIDS Reagents. The following reagents were obtained from the AIDS Research and Reference Reagent Program, Division of AIDS, NIAID: AMD3100, Maraviroc, and T20.

### Cell culture

HEK 293T and Jurkat cell lines were obtained and cultured as described [45,56]. PBMCs were isolated from blood of healthy donors by density gradient sedimentation on Histopaque

(Sigma) or Human Pancoll (Pan Biotech). CD4+ T lymphocytes and monocytes were purified from PBMCs using a CD4- or CD14-positive selection kit (CD4 or CD14 microbeads; Miltenyi) according to manufacturer's guidelines. For infection assays, purified CD4+ T cells were maintained for two days in complete RPMI 1640 medium containing recombinant interleukin-2 (IL-2) (300 IU/ml) and phytohemagglutinin (1 μg/ml, Thermo Fisher Scientific), followed by 3 days in the presence of IL-2 alone. Monocytes were differentiated into macrophages for 8 days in RPMI 1640 culture medium supplemented with 10% FCS, antibiotics, and 25 ng/ml of granulocyte-macrophage colony-stimulating factor (GM-CSF) and macrophage colony-stimulating factor (M-CSF) (Miltenyi).

## Virus production and titration

Production of HIV-1 Env-pseudotyped viruses, containing or not BlaM-vpr, in HEK 293T cells was performed as described previously [31,56]. Briefly, 48 h post-transfection of cells with the pNL-SacII-lacZ/env-Ren proviral vector with or without the pCMV-BlaM-vpr plasmid and the pAdvantage vector, culture supernatants were cleared by centrifugation at low speed. BlaM-vpr viruses were further concentrated by ultracentrifugation (72,000 g for 90 min at 4˚C) and then resuspended in DMEM. Virus production was quantified by determining the amount of the Gag p24 protein using an enzyme-linked immunosorbent (ELISA) assay (Innogenetics or TaKaRa). For production of HIV-1 Env-pseudotyped viruses in Jurkat cells, we initially produced VSVg-pseudotyped viruses in HEK 293T cells by cotransfection of the pNL-SacII-lacZ/env-Ren proviral vector in combination with pVSVg using the calcium phosphate precipitation technique [98] or PEI [56]. Virus production in the cell-culture supernatant was then quantified as above and, in some cases, titrated on Jurkat cells using flow cytometry, as described [45]. VSVg-pseudotyped viruses were then used to inoculate Jurkat cells for 24 h (200 ng Gag p24/$10^6$ cells, $2x10^6$ cells/ml), which were then washed twice with phosphate-buffered saline (PBS) before being cultured for another 48 h. Viruses released in the cell-culture supernatant were then collected and measured for their p24 content. For production of viruses dedicated to the cell-free infection experiments conducted in parallel with the cell-to-cell transfer experiments (see below), VSVg-pseudotyped viruses were used at a multiplicity of infection (MOI) of 0.5 to infect Jurkat cells for 16 h. After washing, cells were cultured for another 24 h, and the amount of Gag p24 produced in the cell-culture supernatant was determined by ELISA.

## Cell-free viral infection experiments

Infectivity of cell-free HIV-1 Env-pseudotyped viruses in primary CD4+ T lymphocytes was determined in round bottom 96-well plates. Activated CD4+ T cells ($2x10^5$ /well) in complete RPMI medium supplemented with IL-2 (20 ng/ml) were incubated for 48 h at 37˚C with varying Gag p24 concentrations of viruses (typically ranging between 0.01 and 100 ng/well). For infection of MDMs, $2x10^5$ cells seeded in flat bottom 96-well plates were infected with equal infectious doses of viruses (typically 35,000 or 100,000 RLU of luciferase activity in primary CD4+ T cells) for 48 h at 37˚C. Cells were then washed in PBS and infectivity was determined by measuring *Renilla* luciferase activity in the lysates of infected cells (Promega, Madison, WI, USA) using a Glomax luminometer (Promega).

## Virus-MDM fusion assay

Experiments were performed as previously described [31]. Briefly, $2x10^5$ MDMs (previously collected with cell dissociation buffer, Thermo Fisher Scientific) were exposed to equivalent amounts of infectious virus particles (determined by titration on primary CD4+ T

lymphocytes), in the presence or absence of AMD3100 or MVC (10 μM each), by spinoculation at 2,000 x g for 1 h at 4˚C in 96-well, conical-bottom plates. The virus-cell mixtures were then warmed to 37˚C for 2 h. Cells were then washed twice at room temperature and further incubated with the CCF2/AM dye for 2 h at 37˚C according to the manufacturer's instructions (Invitrogen). Cells were subsequently fixed with 1% paraformaldehyde (PFA) in PBS for 20 min at 4˚C. The percentage of cells with CCF2 cleaved by BLaM was quantified by flow cytometry using the BD LSR-Fortessa (BD Biosciences).

## Cell-to-cell viral transfer to MDMs

Cell-to-cell virus infection assays of MDMs from infected Jurkat cells were carried out as described previously [45]. Briefly, Jurkat cells were initially infected with VSVg-pseudotyped virus stocks produced from HEK 293T cells at a MOI of 0.5 (unless otherwise specified), typically for 16 h and then for another 24 h after an intermediate washing step. In assays on CCR5 and CD4 dependence, Jurkat cells were directly infected for 48 h. After washing, Jurkat cells were cocultured for 6 or 24 h at a 1:1 ratio (or 2:1 ratio in the assays on receptor dependence) with MDMs seeded at a density of 0.5 x $10^6$ cells/well of 12-well cell culture plates. As controls, parallel cell-free infections of 0.5x$10^6$ MDMs with Jurkat cell-derived viruses (250 ng Gag p24) were carried out for 6 or 24 h. Where indicated, MDMs were pretreated before coculture for 1 h with T20 at 10 μg/ml, MVC and/or AMD3100 at the concentrations specified in the figure legends. After coculture, Jurkat cells were eliminated by extensive washes in PBS and 4 mM EDTA-containing PBS [45]. In the case of assays on receptor dependence, MDMs were further incubated for additional 48 h in the presence or absence of entry inhibitors. MDMs were then surface stained with anti-CD11b and, in some instances, with anti-CD4 or anti-CCR5 mAbs, fixed and permeabilized [45,56], and the percentages of Gag+ cells were determined by flow cytometry using the KC57 anti-Gag mAb (1/500), as previously detailed [45]. For measurement of virus infectivity after cell-to-cell viral transfer, MDMs infected as above in the presence of Jurkat cells or with cell-free viruses were cultured for 48 more hours before quantification of the luciferase activity as previously in the cell lysates.

## Fluorescence microscopy

To visualize virus cell-to-cell transfer through cell-cell fusion, infected Jurkat cells were initially preloaded with 5 μM of CellTracker CMAC (7-amino-4-chloromethylcoumarin) (Life Technologies) and cocultured for 6 or 24 h with MDMs plated onto coverslips. Where indicated, MDMs were pretreated before coculture for 1 h with MVC, AMD3100 or T20, as above. After elimination of Jurkat cell by washing, MDMs were directly fixed with 4% PFA, blocked for 10 min in PBS containing 1% bovine serum albumin (BSA) and stained with 2 μM of DRAQ5 (eBioscience) for 20 min in PBS for staining of nuclei. MDMs were then permeabilized and stained using KC57 FITC-conjugated anti-Gag antibody (1/200 dilution) and phalloidin-Alexa Fluor 647 (Molecular Probes) diluted in permeabilization buffer for 1 h. Coverslips were then washed with PBS and mounted on slides using 10 μl of Fluoromount (Sigma). Images were acquired on a spinning disk (CSU-X1M1; Yokogawa)-equipped inverted microscope (DMI6000; Leica) and were then processed using Fiji software (ImageJ; NIH). Quantitative image analysis was performed using Fiji by defining a region of interest using the actin staining and measuring the whole fluorescence intensity of the Gag staining, with respect to noninfected cells. The number of CellTracker+ nuclei, as well as the number of DRAQ5+ nuclei, per MDM were determined from images of at least 100 cells followed by processing using Fiji as described previously [47].

## Statistical analysis

Statistics and curve fitting were performed using the GraphPad prism software.

## Supporting information

**S1 Fig. Cell-to-cell transfer of R5 M- and non-M-tropic HIV-1.** Jurkat cells infected with the indicated virus clones were co-cultured for 24 h with MDMs. After elimination of Jurkat cells, MDMs were stained with anti-Gag, phalloidin (actin) and DRAQ5 (nuclei), and analyzed by confocal microscopy (scale bar, 10 μm). Representative images are shown in **A)**. The total number of nuclei (DRAQ5+) per Gag+ MDM was quantified from images on at least 200 cells. MDMs cocultured with non-infected Jurkat cells were used as negative controls (NI). In **B)**, results are expressed as the percentages of Gag+ MDMs with 1, 2, 3 or more than 3 DRAQ5(+) nuclei. In **C)**, results are expressed as the number of DRAQ5(+) nuclei per Gag+ MDM; each dot corresponds to 1 cell. Horizontal bars represent means +/- 1 SEM. The results shown are representative of at least 6 independent experiments performed with MDMs from at least 6 different donors.
(TIF)

**S2 Fig. Cell-to-cell transfer to macrophages of CXCR4-using HIV-1.** Jurkat cells infected with the indicated viruses were co-cultured for 24 h with MDMs. After elimination of Jurkat cells, MDMs were stained with anti-Gag, phalloidin (Actin) and DRAQ5 (nuclei), and analyzed by confocal microscopy (scale bar, 10 μm). Representative images are shown in **A)**. The total number of nuclei (DRAQ5+) per Gag+ MDM was quantified from images on at least 100 cells. MDMs cocultured with non-infected Jurkat cells were used as negative controls (NI). In **B)**, results are expressed as the percentages of Gag+ MDMs with 1, 2, 3 or more than 3 DRAQ5(+) nuclei. In **C)**, results are expressed as the number of DRAQ5(+) nuclei per Gag + MDM; each dot corresponds to 1 cell. Horizontal bars represent means +/- 1 SEM. The results shown are representative of at least 6 independent experiments performed with MDMs from at least 6 different donors.
(TIF)

**S3 Fig. Dual (R5X4) tropism of 89.6 and X4-1 Envs in cell-to-cell viral transfer between infected T cells and MDMs.** Jurkat cells were infected with X4-1 (**A-C**) or 89.6 (**D-F**) Env-pseudotyped viral clones, and then cocultured for 24 h with MDMs pretreated or not (mock) with AMD3100, MVC or both. After elimination of Jurkat cells, MDMs were stained with anti-Gag, phalloidin (Actin) and DRAQ5, and analyzed by confocal microscopy (scale bar, 10 μm). Representative images are shown in **A)** and **D**). The total number of nuclei (DRAQ5 +) per Gag+ MDM was quantified from images on at least 100 cells. MDMs cocultured with non-infected Jurkat cells were used as negative controls (NI). In **B)** and **E)**, results are expressed as the percentages of Gag+ MDMs with 1, 2, 3 or more than 3 DRAQ5(+) nuclei. In **C)** and **F)**, results are expressed as the number of DRAQ5(+) nuclei per Gag+ MDM cocultured with infected Jurkat cells; each dot corresponds to 1 cell. Horizontal bars represent means +/- 1 SEM, and statistical significance was determined with the Mann-Whitney U-test (**, $P < 0.01$; ****, $P < 0.0001$). The results shown are representative of at least 4 independent experiments performed with MDMs from at least 4 different donors.
(TIF)

**S4 Fig. Analysis of Env incorporation into viruses produced in Jurkat or HEK 293T cells. A)** Western blot analysis of gp120 and p24 expression into viruses, Jurkat or HEK 293T-derived, pseudotyped with JR-FL Env. For each virus, 130 ng of Gag p24 were solubilized in

lysis buffer containing XT sample buffer (Biorad), Invitrogen NuPAGE sample reducing agent and 1% Triton X-100, incubated for 5 min at 70˚C, loaded onto Biorad Criterion XT 4–12% Bis-Tris gels under reducing conditions and then transferred to nitrocellulose membrane. Membranes were blocked with Odyssey blocking buffer (Li-COR) (for p24 detection) or TBS containing 5% BSA and 0.05% NaN3 (for gp120 detection) and then incubated overnight at 4˚C with a sheep anti-HIV-1 gp120 polyclonal antibody (clone D7324, Aalto Bio Reagents) or for 1 h at RT with a mouse anti-HIV-1 p24 mAb (clone 749140, R&D Systems). Membranes were incubated with the following species-specific secondary antibodies: DyLight 800-conjugated donkey Anti-Sheep IgG (Novusbio) and IRDye 800CW-conjugated goat Anti-Mouse (Li-COR) (dilution: 1/10,000). Signals were detected with a Li-COR Odyssey scanner and quantified using ImageStudioLite software. Arrow indicates gp120 bands. Results from two independent experiments with two distinct virus productions, carried out in duplicate, are shown. **B)** Band intensity ratios (means ± SD) of gp120 to p24. Statistics: Mann-Whitney U-test.
(TIF)

**S5 Fig. Cell-to-cell transfer of T/F viruses from infected CD4+ T cells to MDMs or dendritic cells. A)** Blood primary CD4+ T cells were purified, infected with the indicated T/F viruses, and then cocultured with autologous MDMs for 24 h. MDMs were then fixed and stained with phalloidin (Actin), anti-Gag and Dapi (Nuclei), and analyzed by confocal microscopy. Representative images are shown. **B)** and **C)** The total number of nuclei (Dapi+) per Gag + MDM was quantified from images on at least 30 cells. MDMs cocultured with non-infected CD4+ T cells were used as negative controls (NI). In **B)**, results are expressed as the number of Dapi(+) nuclei per Gag+ MDM; each dot corresponds to 1 cell. Statistical significance was determined with the Mann-Whitney U-test (ns, not significant, P > 0.05; ****, P < 0.0001). In **C)**, results are expressed as the means of total nucleus number (Dapi+) per Gag+ MDM. The results represent means of 2 independent experiments performed with MDMs of 2 different donors. Horizontal bars represent means +/- 1 SEM. **D)** Blood monocytes were differentiated into immature DCs for 5 days with GM-CSF and IL-4 and then cocultured for 6 h with Jurkat cells infected with the indicated T/F viruses. The percentage of CD11c+/Gag+ DCs was determined by flow cytometry after intracellular Gag staining. DCs cocultured with non-infected Jurkat cells were used as negative controls (NI). **E)**, Jurkat cells infected with THRO and REJO viruses and prelabeled with CellTracker were cocultured for 6 h with DCs. DCs were then stained with anti-Gag, anti-DC-SIGN and DRAQ5 (nuclei), and analyzed by confocal microscopy (scale bar, 10 μm).
(TIF)

## Acknowledgments

We thank Christel Verollet, Brigitte Raynaud-Messina, Nikaïa Smith and Jean-Philippe-Herbeuval for constant scientific discussions and advices, as well as editing of the final manuscript.

## Author Contributions

**Conceptualization:** Mingyu Han, Vincent Cantaloube-Ferrieu, Maorong Xie, Marie Armani-Tourret, Jean-Christophe Pagès, Philippe Colin, Bernard Lagane, Serge Benichou.

**Data curation:** Mingyu Han, Vincent Cantaloube-Ferrieu, Maorong Xie, Marie Armani-Tourret, Marie Woottum, Jean-Christophe Pagès, Philippe Colin, Bernard Lagane, Serge Benichou.

**Formal analysis:** Mingyu Han, Vincent Cantaloube-Ferrieu, Maorong Xie, Marie Armani-Tourret, Jean-Christophe Pagès, Bernard Lagane, Serge Benichou.

**Funding acquisition:** Bernard Lagane, Serge Benichou.

**Investigation:** Maorong Xie, Bernard Lagane, Serge Benichou.

**Methodology:** Mingyu Han, Vincent Cantaloube-Ferrieu, Maorong Xie, Marie Armani-Tourret, Marie Woottum, Bernard Lagane, Serge Benichou.

**Project administration:** Serge Benichou.

**Resources:** Bernard Lagane, Serge Benichou.

**Supervision:** Jean-Christophe Pagès, Bernard Lagane, Serge Benichou.

**Validation:** Mingyu Han, Vincent Cantaloube-Ferrieu, Maorong Xie, Marie Armani-Tourret, Jean-Christophe Pagès, Philippe Colin, Bernard Lagane, Serge Benichou.

**Visualization:** Maorong Xie, Bernard Lagane, Serge Benichou.

**Writing – original draft:** Bernard Lagane, Serge Benichou.

**Writing – review & editing:** Mingyu Han, Vincent Cantaloube-Ferrieu, Maorong Xie, Marie Armani-Tourret, Marie Woottum, Jean-Christophe Pagès, Philippe Colin, Bernard Lagane, Serge Benichou.

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
