## [Decision Letter · Decision Letter 0]

15 Mar 2022

Dear Dr. Benichou,

Thank you very much for submitting your manuscript "HIV-1 cell-to-cell spread overcomes the virus entry block of non-macrophage-tropic strains in macrophages" for consideration at PLOS Pathogens. As with all papers reviewed by the journal, your manuscript was reviewed by members of the editorial board and by several independent reviewers. The reviewers appreciated the attention to an important topic. Based on the reviews, we are likely to accept this manuscript for publication, providing that you modify the manuscript according to the review recommendations.

Sincerely,

Guido Silvestri

Associate Editor

PLOS Pathogens

Susan Ross

Section Editor

PLOS Pathogens

Kasturi Haldar

Editor-in-Chief

PLOS Pathogens

orcid.org/0000-0001-5065-158X

Michael Malim

Editor-in-Chief

PLOS Pathogens

orcid.org/0000-0002-7699-2064

Reviewer Comments (if any, and for reference):

Reviewer's Responses to Questions

**Part I - Summary**

Reviewer #1: an et al. analyzed infection of macrophages via cell free and cell-associated HIv-1 strains and found that macrophages are readily infectable via cell-to-cell transfer but not cell-free virus. These findings suggest that HIV-1 has a more prevalent tropism for macrophages as initially thought (and suggested by standard in vitro infection assays with cell-free virus) and may impact our understanding of the role macrophages in HIV-1 pathogenesis and/or persistence. The experiments were mostly performed in an established cell-to-cell assay including important controls such as specific X4 and R5 inhibitors and included a panel of previously characterized HIV-1 strains. The data are described in great detail and highly interesting for the field. I support publication, perhaps the following minor aspects could be clarified:

Reviewer #2: In the manuscript entitled “HIV-1 cell-to-cell spread overcomes the virus entry block of non-macrophage-tropic strains in macrophages” Han and colleagues have investigated the susceptibility of human monocyte-derived macrophages (MDM) to infection by “macrophage-tropic” vs. “T-cell tropic” HIV-1 by comparing cell-free vs. cell-to-cell viral infection from the CD4+ Jurkat T cell line to MDM. In particular, they investigated the capacity of different CCR5-dependent (R5) and CXCR4-dependent (X4) HIV-1 by using viruses expressing Env including those fro transmitter/founder (T/F) viruses. The authors observed that most viruses were unable to infect MDM as free virions, whereas all viruses could be efficiently transmitted from T cells to MDM by cell-to-cell contact via Env-dependent fusion of the plasma membranes leading to formation of hybrid multinucleate giant cells (MGC). This was associated with an increased efficiency of both CD4 and CCR5 use although not related to a decreased susceptibility to the inhibitory effects of maraviroc, a CCR5 antagonist. Of interest, the authors report a different modality of transfer of T/F viruses from infected T cells to MDM, i.e. by gp120 Env-mediated cell-to cell fusion leading to MGC formation and by inclusion (likely by phagocytosis) of infected cells without formation of MGC. The authors also describe a partial change in Co-R use in the case of the X4-1 Env expressing virus in that it showed susceptibility to inhibition by MVC plus AMD3100 in cell-cell infection experiments, while it was a pure CXCR4-dependent virus by cell-free infection.

Reviewer #3: Han et al. present data characterizing the infection of macrophages by HIV-1 in vitro, supporting a model in which infected T cells induce productive infection of macrophages regardless of the macrophage tropism of the viruses apparent when cell-free viruses are used as the inocula. The cell-cell mode of infection depends on CD4 and co-receptors (CCR5 or CXCR4), and is associated with cell-cell fusion as shown by the uptake of T cell nuclei into the macrophages. The co-receptor usage of the each of the viral Env proteins tested is usually the same in either mode of infection, but the cell-cell mode is somewhat less sensitive to receptor-blockers and seems to require lower levels of receptors. Much of the data depends on viral pseudotypes, but the study concludes with the use of replication-competent transmitted/founder viruses, some of which (2 of 4) show the same effect.

Overall, this study extends previous work on the infection of macrophages by T cells by showing that this process occurs for nominally non-M-tropic viruses including CXCR4-tropic and transmitted/founder viruses and that it occurs by cell-cell fusion in a CD4- and co-receptor dependent manner. The study is carefully and rigorously done with attention to replicates and the use of multiple donors for the primary macrophages.

**Part II – Major Issues: Key Experiments Required for Acceptance**

Reviewer #1: (No Response)

Reviewer #2: (No Response)

Reviewer #3: 1) The study exclusively uses the Jurkat T cell line as the "donor" in the T cell- macrophage infections. It seems important to confirm the mechanism using primary CD4-positive T cells, perhaps using the two transmitted/founder viruses that show it, to maximize the relevance.

**Part III – Minor Issues: Editorial and Data Presentation Modifications**

Reviewer #1: I support publication, perhaps the following minor aspects could be clarified:

Line 117: 293- affinofile? Meaning?

Line 160: Are used NL43 reporter viruses nef defect and how does this affect viral infectivity and readout?

Fig. 1A/B. Do results correlate (infectivity in CD4TL vs macrophages)? Same for 1C/D and also (as control) 1A with 1C or 1B with 1D

line 365 obtained

Authors discussed several mechanism for why cell-cell transfer is more efficient than cell free infection. Authors should also consider that an increased concentration of virions produced by the donor cell may also enhance infection. This could be mimicked in a cell-free infection experiment by adding concentrated virus in a very small volume to macrophages followed by a spinoculation. Would this allow infection rates comparable to those via cell-cell transfer?

Reviewer #2: The paper is a completion of previous published observations by the authors, it is very interesting and well-written. The interpretation of the results is consistent with the literature and coherent. A few issues should be clarified in order to improve the overall clarity of the paper.

• The technical detail that Jurkat cells were initially infected with VSV-g pseudotyped viruses should be specified in the Results and not only in the Method section.

• In the co-culture experiments, “Jurkat cells were removed by extensive washes” (line 217). I assume that this protocol implies that MDM were adherent to the plastic surface while Jurkat cells were in suspension. This (or an alternative explanation) should be specified in the Results.

Minor points

• The sentence “Therefore, M-tropic viruses should be viewed as more likely to infect MDMs as compared to non-M-tropic viruses” (lines 179-181) is a tautology and should be omitted.

• In relationship to the hypothesis described in lines 307-308, the paper by Massari et al. (“In vivo T lymphocyte origin of macrophage-tropic strains of HIV. Role of monocytes during in vitro isolation and in vivo infection” by F. Massari et al., J Immunol, 1990 Jun 15;144(12):4628-32) should cited.

• Concerning the experimental results described in lines 332-340, an appropriate control would have been to incubate HEK cells with Jurkat-derived supernatant to see whether some inhibitory activity could be detected. In addition, viruses generated from primary T cells, rather than a cell line, should have been tested for their capacity to infect MDM. Perhaps the author could consider these observations expanding the discussion of the results.

Reviewer #3: 2) Two of the transmitted/founder viruses (CHO58 and RPHA) that score positively in the FACS assay for Gag after cell-cell infection seem microscopically to have only been taken up as virus particles (puncta vs. the diffuse staining consistent with synthesis of new Gag protein) and do not seem to have induced multinucleated cells or cells that contain T cell nuclei. The authors should clarify that that is not likely to represent productive infection. They could choose to document this by continuing the experiment for several days; presumably viral spread in the culture would not be evident for these.

3) Figure 7 panel H: Are the label for AMD and MVC incorrect (switched)? It appears that AMD is inhibiting formation of multinucleated cells rather than MVC, which is not as stated in the text of the Results and would not make sense.

4) Legend of Figure 3 could restate the meaning of the grey bar in panels A and B (presumably same as in Figure 2).

5) Line 360: "inverse agonist" Is "antagonist" meant?

PLOS authors have the option to publish the peer review history of their article (what does this mean?). If published, this will include your full peer review and any attached files.

Reviewer #1: No

Reviewer #2: **Yes: **Guido Poli

Reviewer #3: No

Figure Files:

Data Requirements:

Reproducibility:

References:

---

## [Editor Report · Decision Letter 1]

9 May 2022

Dear Dr. Benichou,

We are pleased to inform you that your manuscript 'HIV-1 cell-to-cell spread overcomes the virus entry block of non-macrophage-tropic strains in macrophages' has been provisionally accepted for publication in PLOS Pathogens.

Best regards,

Guido Silvestri

Associate Editor

PLOS Pathogens

Susan Ross

Section Editor

PLOS Pathogens

Kasturi Haldar

Editor-in-Chief

PLOS Pathogens

orcid.org/0000-0001-5065-158X

Michael Malim

Editor-in-Chief

PLOS Pathogens

orcid.org/0000-0002-7699-2064
---

## [Editor Report · Acceptance letter]

25 May 2022

Dear Dr. Benichou,

We are delighted to inform you that your manuscript, "HIV-1 cell-to-cell spread overcomes the virus entry block of non-macrophage-tropic strains in macrophages," has been formally accepted for publication in PLOS Pathogens.

Best regards,

Kasturi Haldar

Editor-in-Chief

PLOS Pathogens

orcid.org/0000-0001-5065-158X

Michael Malim

Editor-in-Chief

PLOS Pathogens

orcid.org/0000-0002-7699-2064